# CONCEPTUAL GRAPH COUNTERFACTUALS

## ABSTRACT

Conceptual counterfactuals refer to hypothetical scenarios involving changes in a high-level conceptual representation. In the realm of XAI, conceptual Counterfactual Explanations (CEs) allow for more meaningful and interpretable modifications. For instance, instead of explaining image predictions through superficial pixel-level changes, the focus shifts to alterations in the underlying semantics. In this work, we propose representing input data as semantic graphs to achieve more descriptive, accurate, and human-aligned explanations. Furthermore, we introduce a model-agnostic GNN-powered method to efficiently compute counterfactuals. We begin by representing images as scene graphs and obtain appropriate representations through GNNs to bypass solving the NP-hard graph similarity problem for all input pairs, an integral part of the CE computation process. We apply our method to widely-used datasets and compare our CEs with previous state-of-the-art explanation models based on semantics, including both white and black-box approaches. We outperform both approaches quantitatively and qualitatively, as validated by human subjects, specifically when the graphs contain numerous edges, highlighting the significance of capturing intricate relationships. Given the model-agnostic nature of our approach and the generalizability of the graph representation, this method is successfully extended to diverse modalities and classifiers, including non-neural models. Additionally, it is proven consistent across generated annotations, at least in the case of scene graph generation. Our approach is, to our knowledge, the first to emphasize semantic graphs as a vehicle for CEs, allowing the transition from low-level features to concepts. It uniquely leverages graph matching GNNs as a XAI tool achieving efficient approximation and significant acceleration in comparison to the exact Graph Edit Distance (GED) algorithm. It is widely applicable and easily extensible, producing actionable explanations.

## 1 INTRODUCTION

The pervasiveness of deep learning applications combined with our lack of knowledge about the inner workings of black-box AI systems has recently brought the importance of eXplainable AI (XAI) to a critical juncture Arrieta et al. (2020). Cultivating trust between humans and machines necessitates that individuals can comprehend and exert control over opaque systems, if not avoid them completely Rudin (2019). One of the most instinctive forms of reasoning triggered in humans is counterfactual thinking, i.e. imagining alternative scenarios that would have led to a change of outcome. To this end, many recent XAI ventures are centered upon counterfactuals given their informative nature and accessibility Guidotti (2022). Despite widespread human adoption of the provided explanation being a desirable characteristic, it is insufficient in itself. Focusing on visual classifiers, many previous feature attribution-based works emphasize finding, highlighting, and modifying impactful areas of an image that lead to a different classification result, in a counterfactual manner (Goyal et al., 2019; Zhao et al., 2021; Vandenhende et al., 2022; Augustin et al., 2022) or not (Ribeiro et al., 2016; Selvaraju et al., 2017; Adebayo et al., 2018). No matter how attractive these approaches may seem, they often do not showcase the whole truth. For example, marking just pixels corresponding to a bird's wing to justify class prediction as in Goyal et al. (2019) or Vandenhende et al. (2022), not only disregards the existence of numerous other differences between bird species but also does not meaningfully explain what edits need to be made to guarantee the transition to the other class.

In the interest of more complete, realistic, and actionable CEs, we leverage a high-level representation of input data. Following the claim that 'there is no explanation without semantics', as has been theoretically and experimentally proven (Browne & Swift, 2020),

we base our method on a semantic standpoint, avoiding the task of assigning meaning to individual layers of a neural network or isolated pixel areas within images. Building upon recent works that represent images as sets of concepts or attributes (Filandrianos et al., 2022; Abid et al., 2022; Dervakos et al., 2023), we go one step further and additionally model relations between depicted concepts as a graph. For instance, in Fig 1 (top) we present the graph corresponding to a Parakeet Auklet, which consists of bird parts and their respective attributes. The graph representation enables us to tap into the inherent structure of the data and preserve valuable semantic information. These semantics can align with external ontological knowledge, establishing that concepts such as 'white' and 'black' as colors are more closely related to descriptors like 'striped' than to specific bird parts like 'leg'. The general nature of the graph structure makes this framework applicable to any modality with accompanying annotations and thus extensible to corresponding predictors, i.e. audio classifiers. Despite its dependence on annotations, our method maintains consistency in drawing global conclusions, regardless of the annotation tool.

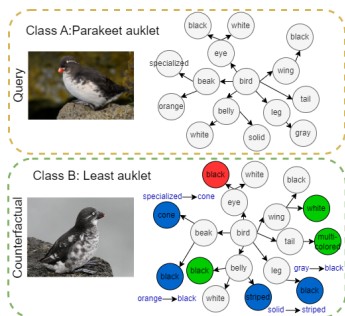

Figure 1: Graph edits as CEs for predicted classes A and B. Colored nodes denote the *minimum* number of concepts to be inserted, deleted or substituted to perform the A →B transition. Edge types are omitted.

In terms of our CE process, similar to the Semantic Counterfactuals (SC) of Dervakos et al. (2023), we focus on counterfactual retrieval of the most semantically similar instance in the target class but define similarity through graph edit distance (GED) computation. Once the counterfactual image is obtained, the explanation can be served to humans as in Fig. 1 (bottom graph). Since CEs are sampled from the existing data distribution and calculated considering all necessary edits from query to target image, our explanations are always actionable. Despite its merits, however, GED does belong in the NP-hard complexity class. To this end, we offer building a graph similarity Graph Neural Network (GNN) model to approximate it; thus effectively computing it using only about half of the input pairs and ultimately *calculating only one edit path* per query image after retrieval. Finally, the proposed method is black-box, eliminating the need for peeking inside the classifier. The versatility of our framework is a significant advantage, as it allows for post-hoc explanations of virtually *any* model capable of labeling instances, without compromising its performance when explaining a specific model.

To summarize, the contributions of our method can be grouped as such: a) more **interpretable and semantically meaningful** CEs because of semantic graphs, b) **increased flexibility** due to its model-agnostic nature, combined with competence against other black- and white-box techniques, c) **improved efficiency** through GNN approximation of GED, and d) **actionability**. Our approach is the first to employ graphs and GNNs for counterfactual retrieval; presented results are assessed across four diverse datasets (images and audio) with three neural classifiers and one experiment on explaining human decision, involving two human surveys, and four experiments with quantitative and qualitative guarantees, achieving superior performance and runtime compared to SOTA algorithms.

## 2    RELATED WORK

**Counterfactual explanations**    In visual classifier interpretability, recent research has increasingly favored counterfactual approaches. Pixel-level edit methods focus on marking and altering significant image areas to influence the model's predictions (Goyal et al., 2019; Vandenhende et al., 2022; Augustin et al., 2022); some even leveraging advanced generative techniques. Contrary to other feature extraction counterfactual methods, the Counterfactual Visual Explanations (CVE) of Vandenhende et al. (2022) attempt to enforce semantically consistent area exchanges through an auxiliary model for semantic similarity prediction between local regions. Their semantics-centered approach lies closest to ours, which is geared towards human-interpretable concept edits (Filandrianos et al., 2022; Abid et al., 2022; Dervakos et al., 2023). Abid et al. (2022) propose conceptual CEs in the event of a misclassification. Their method requires white-box access to the model and therefore suffers in the same way as the majority of visual CEs. In contrast, our approach emphasizes a model-agnostic perspective. We adopt the definition of concepts as objects or relations linked with ontological knowledge, semantically enriching the model as in previous works (Alirezaie et al., 2018; Zhao et al., 2021). In contrast to Filandrianos et al. (2022) who leverage Set Edit Distance as a proxy, ignoring

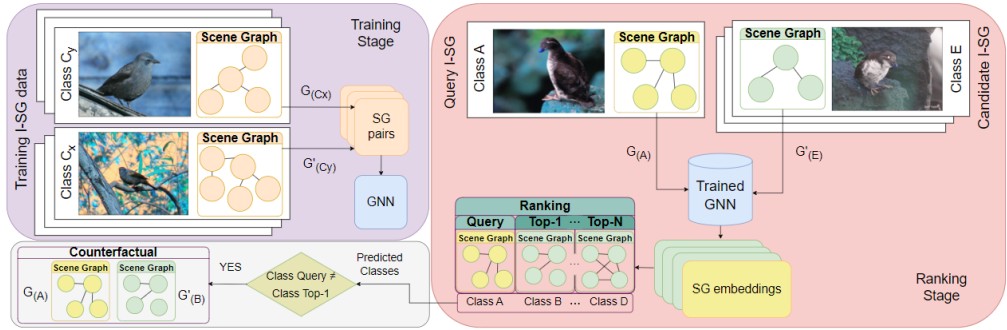

Figure 2: Outline of our method (for image classifiers). The depicted stages directly correspond to Sec. 3 paragraphs. Class labels predicted by given classifier are: A - query, B - target, $C_x$, $C_y$ - any class, others - random class instances. Graph $\mathcal{G}'_{(B)}$ corresponds to counterfactual image $I'_{(B)}$.

edges of the semantic graph, and SC (Dervakos et al., 2023) who extend the previous approach by rolling up the edges into concepts, sacrificing crucial object relation information, we use the more accurate GED. We build upon SC by introducing semantic graphs for increased expressivity, and use GNNs for the acceleration of GED calculation. While our method is also domain-agnostic, this paper centers on the visual domain and later extends to other modalities. Finally, our approach has the key novelty that it harnesses the capabilities of GNNs to provide CEs, in contrast to existing literature that focuses on the inverse task of explaining GNNs themselves (Bajaj et al., 2021; Lucic et al., 2022).

**Graph similarity**   Given the computational complexity of methods such as graph edit distance (GED) (Sanfeliu & Fu, 1983), using approximation algorithms is a practical choice. Fankhauser et al. (2011) introduced a GED approximation that combines the Jonker-Volgenant assignment algorithm with a bipartite heuristic leading to significant speedup, which we adopt for the fast computation of GED during training. Considering neural approaches, the ones relevant to our work leverage GNNs (Bai et al., 2019a;b; Li et al., 2019; Ranjan et al., 2022). These techniques commonly involve training two identical GNNs using graph pairs as input and their similarity to compute loss. As our paper focuses on embedding extraction to facilitate CEs instead of the similarity itself, we draw inspiration from previous approaches in implementing our GNN model, by adopting the ideas of Siamese GNNs, graph-to-graph proximity training and Multi-Dimensional Scaling as loss (Bai et al., 2019b) to preserve inter-graph distances in the embedding space.

## 3   METHOD

Since the majority of our experiments are conducted with visual classifiers, we will illustrate our framework within this domain. Given a query image $I_{(A)}$ belonging to a class $A$, a conceptual CE entails finding another image $I'_{(B)} \neq I_{(A)}$ belonging to a class $B \neq A$, so that the shortest edit path between $I_{(A)}$ and $I'_{(B)}$ is minimized. Even though there are different notions of distance between images, we select a *conceptual* representation, employing scene graphs to represent objects and interactions within images. To this end, the problem of image similarity ultimately reduces to a graph similarity challenge. However, graph edits (insertions, deletions, substitutions) as a deterministic measure of similarity between two graphs $\mathcal{G}_{(A)}$ and $\mathcal{G}'_{(B)}$ is an NP-hard problem. Optimal edit paths can be found through tree search algorithms with the requirement of exponential time. When searching for a counterfactual graph to $\mathcal{G}_{(A)}$ among a set of $N$ graphs, GED needs to be calculated $N - 1$ times. To minimize the computational burden, we use lightweight GNN architectures that accelerate the graph proximity procedure by mapping all $N$ graphs to the same embedding space. By retrieving the closest embedding to $\mathcal{G}_{(A)}$ that belongs to a different class $B \neq A$, GED should be computed only *once* per query during the retrieval stage. Concretely, we approximate the following optimization problem for semantic graphs extracted from any input modality:

$$GED(min|\mathcal{G}_{(A)}, \mathcal{G}'_{(B)}|), \text{ such that } A \neq B \qquad (1)$$

**Ground Truth Construction**   As our overall approach does not rely on pre-annotated graph distances and semantic relationships, we propose a technique to construct well-defined ground truth

instances. The graph structure of data imposes the requirement of defining an absolute similarity metric between graph pairs for the training stage. GED is regarded as the optimal choice despite its computational complexity; computing GED for only $N/2$ pairs to construct the training set is adequate for achieving high quality representations, as validated experimentally. To further facilitate GED calculation, we exploit a suboptimal algorithm utilizing a bipartite heuristic that accelerates an already effective in practice LSAP-based algorithm for GED (Jonker & Volgenant, 1987; Fankhauser et al., 2011). Consequently, semantic information of nodes and edges should guide graph edits based on their conceptual similarity. Thus, we choose to deploy the technique proposed in SC (Dervakos et al., 2023) to assign operation costs based on conceptual edit distance, as instructed by the shortest path between two concepts within the WordNet hierarchy (Miller, 1995).

**GNN Training**  To accelerate the retrieval of the most similar graph $\mathcal{G}'_{(B)}$ to graph $\mathcal{G}_{(A)}$, we build a siamese GNN component for graph embedding extraction based on inter-graph proximity. The GNN comprises two identical node embedding units that receive a random graph pair $(\mathcal{G}_{(C_x)}, \mathcal{G}'_{(C_y)})$ as input ($C_x, C_y$ can be any class). The extracted node representations are pooled to produce global graph embeddings $(\boldsymbol{h}_{\mathcal{G}_{(C_x)}}, \boldsymbol{h}_{\mathcal{G}'_{(C_y)}})$. Embedding units consist of stacked GNN layers, described by either GCN (Kipf & Welling, 2016), GAT (Veličković et al., 2017) or GIN (Xu et al., 2018). We formalize GCN graph embedding computation in Eq. 2 (omitting class notation for simplicity):

$$\boldsymbol{h}_{\mathcal{G}} = \frac{1}{n} \sum_{i=1}^{n} (\boldsymbol{u}_i^{K-1} + \sum_{j \in \mathcal{N}(i)} \boldsymbol{u}_j^{K-1}) \tag{2}$$

where $\boldsymbol{u}_i$ is the representation of node $i$, $\mathcal{N}(i)$ is the neighborhood of $i$, $n$ is the number of nodes for $\mathcal{G}$ and $K$ is the number of GCN layers. To preserve the similarity of vectors $(\boldsymbol{h}_{\mathcal{G}_{(C_x)}}, \boldsymbol{h}_{\mathcal{G}'_{(C_y)}})$, we adopt the dimensionality reduction technique of Multi-Dimensional Scaling (Williams, 2000), as proposed in Bai et al. (2019b). The model is trained transductively to minimize the loss function $\mathcal{L}$:

$$\mathcal{L} = \mathbb{E}(\left\| (\boldsymbol{h}_{\mathcal{G}_{(C_x)}} - \boldsymbol{h}_{\mathcal{G}'_{(C_y)}}) \right\|_2^2 - GED(\mathcal{G}_{(C_x)}, \mathcal{G}'_{(C_y)})) \tag{3}$$

Graphs are embedded in a lower dimensional space by choosing a random subset of $\frac{N!}{2(N-2)!}$ pairs with varying cardinality $p$. The node features initialization is significant with regard to semantic similarity preservation; thus, we use GloVe representations (Pennington et al., 2014) of node labels.

**Ranking Stage and Counterfactual Retrieval**  Once graph embeddings have been extracted, they are compared using cosine similarity as a metric in order to produce rankings. For each query image $I_{(A)}$ and subsequently its scene graph $\mathcal{G}_{(A)}$, we obtain the instance $\mathcal{G}'_{(B)}$ with the highest rank given the constraint that $I'_{(B)}$ belongs in a class $B \neq A$. Image $I'_{(B)}$ is proposed as a counterfactual explanation of $I_{(A)}$ since it constitutes the instance with the minimum graph edit path from it, which is classified in a different target category $B$. Specifically, we retrieve a scene graph $\mathcal{G}'_{(B)}$ as:

$$\mathcal{G}'_{(B)} = \mathcal{G}^i_{(B)}, \ \arg\max_i \left( \frac{\boldsymbol{h}_{\mathcal{G}^i_{(B)}} \cdot \boldsymbol{h}_{\mathcal{G}_{(A)}}}{\left\| \boldsymbol{h}_{\mathcal{G}^i_{(B)}} \right\| \left\| \boldsymbol{h}_{\mathcal{G}_{(A)}} \right\|} \right) if \ A \neq B, \qquad i = 1, ..., N \tag{4}$$

The choice of target class $B$ is strongly correlated with the characteristics of the dataset in use and the goal of the explanation itself. To be precise, if the data instances have ground truth labels, the target class could be defined as the most commonly confused one compared to the source image class, as in Vandenhende et al. (2022). Another valid choice would be to arbitrarily pick $B$ to facilitate a particular application, i.e. explanation of classifier mistakes, in which case $B$ is the true class of the query image, as in Abid et al. (2022). We utilize the first approach when class labels are available; otherwise, when no ground truth classification labels exist, we propose defining the target class as the one with the most highly ranked instance classified differently than source image $I_{(A)}$.

## 4 EXPERIMENTS

In the following experiments, we present results involving $p \sim N/2$ training graph pairs and the GCN variant unless mentioned otherwise. We produce graph representations using a single Tesla K80 GPU, while all other computations are done on CPU. We utilize PyG (Fey & Lenssen, 2019) for the implementation of the GNN and DGL (Wang et al., 2019) for approximate GED label calculation.

**Evaluation** comprises quantitative metrics, as well as human-in-the-loop experiments. Quantitative results are extracted by comparing the ranks retrieved based on our obtained graph embeddings to the ground truth ranks retrieved by GED, thus reporting *average Precision@k (P@k)* and *NDCG@k*. In this case, all top-k GED retrieved results are considered relevant and equally weighted for NDCG computation. We also design two variants of these metrics which exclusively regard the top-1 GED result as relevant and the rest as irrelevant, denoting them as 'binary'. Comparison between different CE methods is achieved through *average number of edits*. Edits are defined as node/edge insertions, deletions, and substitutions with different concepts. To ensure a fair comparison, edits for all methods are calculated post-hoc through GED, as it provides a complete and comprehensive definition of edit distance. This is necessary because direct comparison with pixel-based methods is not feasible due to the different definitions of units of information (significant rectangular areas vs concepts).

The human evaluation highlights several aspects of our contributions. First, comparison with prior state-of-the-art is necessary to validate the quality of our retrieved CEs. To this end, we ask our evaluators to select among two counterfactual alternatives of a query image. Those alternatives involve an image retrieved from our framework versus an image retrieved either by SC (Dervakos et al., 2023) or CVE (Vandenhende et al., 2022). Moreover, we evaluate the understandability of our CEs by replicating the machine-teaching human experiment of CVE, adjusted to accommodate our graph-based explanations: we design three teaching stages, pre-learning, learning, and testing, and we equally divide our annotators into two independent learning stage variants, namely 'visually-informed' and 'blind'. During pre-learning, we present 10 images and ask annotators to classify them in anonymized classes A and B, or select 'I do not know'. In the 'visually-informed' learning stage, we showcase 10 different counterfactual image pairs retrieved from our method, accompanied by their scene graphs and the graph edits performed to transit from query class A to target class B. On the other hand, the annotators of the 'blind' learning stage are only provided with scene graph pairs and graph edits but no images. This way, we aim to measure the reliance of humans on concepts present on graphs rather than visual information. Finally, in the testing stage, we assess how informative our CEs were for humans in order to discriminate between classes A and B, by evaluating their classification of each image (same as pre-learning). In all cases, our annotators are student volunteers of engineering backgrounds. More details regarding human experiments are provided in Appendix A.

**Experiment objectives** Comparison with CVE showcases the abilities of our model-agnostic approach compared to theirs, which requires white-box model access and relies on pixel-level edits. On the other hand, comparison with SC demonstrates the power of graph representations compared to set-level edits in the black-box conceptual setting. An important clarification is that SC proposes the use of roles only in the corresponding experiments of Sec. 4.3, meaning that for Sec. 4.1, 4.2 they rely solely on depicted concepts. More details are provided in Appendix C, D, E.

## 4.1 COUNTERFACTUALS ON CUB

We apply our approach on the Caltech-UCSD Birds (CUB) (Wah et al., 2011) dataset. CUB does not provide ground truth scene graphs; nonetheless, they can easily be constructed by leveraging given structured annotations. In summary, we create a central node to represent the bird and establish 'has' edges connecting it to all its parts. Each part is then linked to its respective attributes using edges labeled with the corresponding feature type (color, shape, etc.). To achieve consistency with CVE we utilize ResNet50 (He et al., 2015) as the classifier under explanation.

**Quantitative results** We examine the agreement between the counterfactual images $I'_{(B)}$ retrieved by each method (CVE, SC, ours) and the ground truth GED, which serves as the golden standard. Our approach outperforms CVE for every ranking metric (Tab. 2). Regarding SC, metrics are only valid for $k = 1$ since it produces a single CE instead of a rank. Therefore, P@1 for SC is 0.022, much lower than ours. In addition, we observe that our approach leads to the *lowest number of overall edits*: In Tab. 1, we can see that our method produces 10.5 node/edge distortions on average, which is about 1 less edit than SC and 2 fewer edits than CVE, strengthening the claim that our CEs correspond to **minimum-cost edits**.

Table 1: Average number of node, edge & total edits on CUB. **Bold** for best results.

|  | Node↓ | Edge↓ | Total↓ |
|---|---|---|---|
| CVE | 8.43 | 4.70 | 13.13 |
| SC | 8.07 | **3.66** | 11.73 |
| Ours | **6.16** | 4.34 | **10.5** |

Table 2: Comparison of counterfactual retrieval results with ground truth GED rankings on CUB.

| | P@k↑ | | | NCDG@k↑ | | | P@k (binary)↑ | | | NCDG@k (binary)↑ | | |
|---|---|---|---|---|---|---|---|---|---|---|---|---|
| | k=1 | k=2 | k=4 | k=1 | k=2 | k=4 | k=1 | k=2 | k=4 | k=1 | k=2 | k=4 |
| CVE | 0.019 | 0.069 | 0.103 | 0.639 | 0.702 | 0.730 | 0.019 | 0.043 | 0.112 | 0.110 | 0.172 | 0.264 |
| Ours | **0.194** | **0.247** | **0.341** | **0.645** | **0.717** | **0.737** | **0.194** | **0.306** | **0.488** | **0.227** | **0.281** | **0.361** |

**Qualitative results** for CUB are presented in Fig. 3 for three images of class A (Rusty Blackbird), accompanied by the number of edits and GED needed to transition to class B (Brewer Blackbird). Overall, our approach produces the **fewest concept edits**. SC leads to clear fallacies like suggesting CEs with additional birds (SC, left), or with a portion of the bird in view (SC, middle); thus leading to unnecessary costly edits - deletions and additions respectively. In contrast, our model mitigates such errors by utilizing graphs, where concept instances are uniquely tied to nodes, and their interconnections strongly guide graph similarity through GED, ultimately producing a more accurate and expressive notion of distance than flat unstructured sets. CVE generally fails in finding CEs conceptually similar to the query $I_{(A)}$, as highlighted by the elevated number of edits. This pixel-level approach avoids SC's mistakes to an extent by implicitly taking visual features like zoom into account. However, it offers no semantic guarantees, unlike our well-defined GED-based approach.

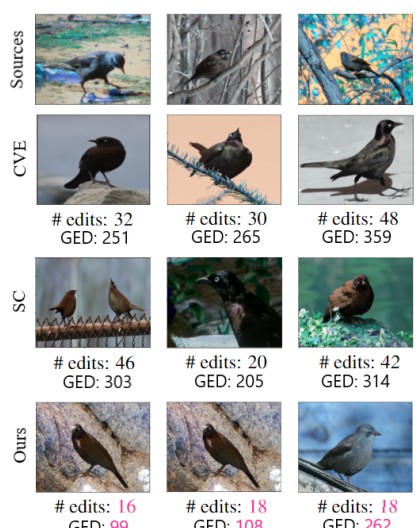

Figure 3: CE results between classes Rusty → Brewer Blackbird. Colored numbers denote best results (lower values).

**Human evaluation** Analyzing the results from the comparative human survey (Tab. 3), we deduce that our CEs are **more human-interpretable** than both SC and CVE by a landslide: annotators prefer our CEs at nearly twice the rate of the CVE alternative. Compared to SC, despite the increased amount of undecided annotators, our CEs were preferred 2.6 times more frequently. This observation proves that despite the closeness of the two concept-based methods, ours is more intuitive to humans, confirming the meaningful addition of linking concepts as part of a graph. As for the machine-teaching experiment, we obtain the test set accuracy scores (Tab. 4), as the ratio of correctly human-classified test images over the total number of test images. Our visually-informed accuracy clearly outperforms reported scores of CVE, highlighting that concept-based CEs are more powerful in guiding humans towards understanding discriminative concepts between counterfactual classes compared to non-conceptual pixel-level CEs. The "blind" results show an expected accuracy decrease compared to the visually-informed one, but still outperform those obtained by CVE. The higher accuracy of concept-based over visual CEs affirms the significance humans place on higher-level features for classification.

Table 3: Human Evaluation preference; Win% = % times our method was preferred, Lose% for vice-versa, Tie% when equally preferred.

| Ours | Win% | Lose% | Tie% |
|---|---|---|---|
| SC | **48.86** | 19.32 | 31.82 |
| CVE | **48.42** | 26.27 | 25.31 |

Table 4: Human test accuracy scores for correct classification of samples in classes A and B.

| Experiment | Test acc.%↑ |
|---|---|
| Ours (visually-informed) | **93.88** |
| Ours (blind) | 89.28 |
| CVE | 82.1 |

**Actionability concerns** CVE may lead to non-actionable CEs, despite training on visual semantic preservation. To illustrate with an example, we observe the following: CVE suggests that the sole addition of a striped pattern in a Gray Catbird's wing is adequate to classify it as a Mockingbird. However, by exhaustively generating all annotated attribute combinations of this new bird instance, we easily find several occurring attribute pairs that are not representative of the Mockingbird class; namely, no other Mockingbird has an eyering head pattern and grey breast color. This strongly contrasts with CEs retrieved by our method. Actionability dictates the prescription of attainable goals

achieved through CEs that accurately represent the underlying data distribution (Poyiadzi et al., 2020). To this end, our approach not only selects CEs drawn from the existing target class distribution but also considers all edits needed to convert query to counterfactual image. Therefore, we formalize a more holistic approach to the distance and path between counterfactual pairs and simultaneously leverage existing relations between depicted objects, both visual (relations on the image) and semantic (relations mapped to WordNet synsets). More details are presented in Appendix E.3.

## 4.2 TOWARDS CONCEPTUAL COUNTERFACTUALS

We focus our analysis on conceptual counterfactuals since the previous sections exhibited the indisputable merits of relevant approaches by outperforming the SOTA pixel-level method of CVE in every aspect. In the interest of extending our method to a less controlled dataset, we employ Visual Genome (VG) (Krishna et al., 2017), a dataset containing over 108k human-annotated scene graphs, describing varying scenes of multiple objects and their in-between interactions. We construct two manageable subsets of 500 scene graphs each, which correspond to ∼125k possible training graph pairs for our GNN models. The first subset is randomly selected and denoted as VG-RANDOM, while the second one is chosen to favor higher graph densities and less isolated nodes, so as to highlight the expressive power of object interconnections. We denote this second subset as VG-DENSE. Details are provided in Appendix B. VG instances lack ground truth classification labels, allowing us to test our counterfactual retrieval method without the hard definition of a specific target class. We assign ground truth labels using a pre-trained Places365 classifier Zhou et al. (2017), and regard as counterfactual classes the closest ones in rank.

**Quantitative Results**  We first compare the average number of edits for our method and SC, as displayed in Tab. 5. Initially, numerical results between the two methods seem similar. However, it is visible that our method exhibits superior performance in sce-

Table 5: Average number of node, edge & total edits on VG.

|  | VG-DENSE | | | VG-RANDOM | | |
|---|---|---|---|---|---|---|
|  | Node↓ | Edge↓ | Total↓ | Node↓ | Edge↓ | Total↓ |
| SC | **4.91** | 7.29 | 12.2 | **12.15** | **7.52** | **19.67** |
| Ours | 4.95 | **7.15** | **12.11** | 12.18 | 7.54 | 19.72 |

narios with dense relations (VG-DENSE) as enforced by the lower number of edge edits compared to SC. Since SC disregards high-level relations (by rolling up roles into their concepts) it returns instances with significantly greater semantic distance, as reinforced by Table 13 in the Appendix.

Regarding CE approximation to ground truth GED, results for our approach are presented in the rows of Tab. 6 denoted as GCN-70K. As for SC, we report that they achieve P@1 scores of 0.246 on VG-DENSE and 0.204 on VG-RANDOM, compared to 0.248 and 0.214 retrieved by our method. We deduce that GED approximation is satisfactory for both methods, with ours taking the lead. The closeness of overall results places great significance in human perception for this experiment; therefore, the following qualitative results demonstrate the advantages of our approach.

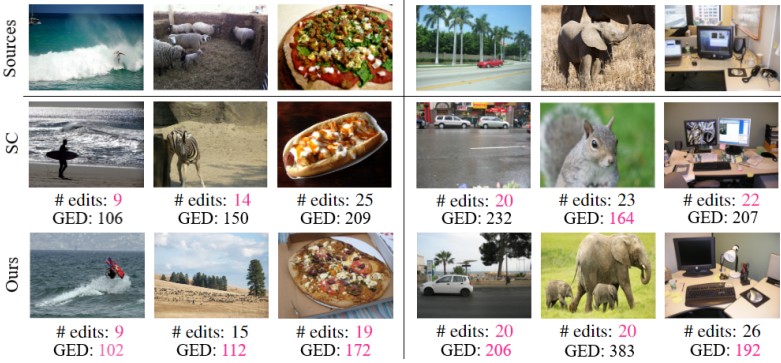

Figure 4: Qualitative results: VG-DENSE (left 3 columns) and VG-RANDOM (right 3 columns).

**Qualitative results**  Through the examination of counterfactual images retrieved for VG-DENSE in Fig. 4(left), there is a clear indication that by considering the complex relations between concepts, our method leads to more **detail-oriented results**. Specifically, in the 1st column, our approach not only

retrieves an image with 'man', 'board', and 'water' concepts, but also containing the relation 'man on board'. In the 3rd column, we take the relation of toppings into consideration and retrieve the pizza, while SC simply retrieves an image with concepts similar in nature, like 'bun' and 'bread' or 'meat' and 'sausage'. Results on VG-RANDOM (Fig. 4 (right)) follow the same logic. In columns 4 and 5, our method retrieves the focal points of the images since it considers relations between palm trees and elephants respectively. Taking into account the sparsity of the underlying graphs, however, in some cases the importance of concepts trumps the underlying structure, as in the 6th column. This fact is reflected in the elevated number of edits of our method for VG-RANDOM, however not always true for GED, showcasing the importance of semantic context. Corresponding graphs are in Appendix E.

**Why GCN?**   Comparison of GNN models in terms of ranking metrics is provided in Tab. 6. Three GNN variants (GAT, GIN, GCN) are trained using $p = N/2 = 70k$ scene graph pairs. The GCN-based counterfactual model consistently approaches GED the closest, with a binary P@4 of 49.0% and P@1 of 24.8% for VG-DENSE and slightly worse results on VG-RANDOM. This ablation study affirms using GCN for the GNN-based similarity component of our approach. We further report that GNNs outperformed other prominent deterministic methods, like graph kernels; results of which can be found in the Appendix D. The reported findings grant us the security that our counterfactual explanations are **trustworthy**, even when applied to complex scene graphs.

Table 6: Ranking results on the two VG variants for different graph-based models.

| Models | P@k ↑ | | | NDCG@k ↑ | | | P@k (binary) ↑ | | | NDCG@k (binary) ↑ | | |
|---|---|---|---|---|---|---|---|---|---|---|---|---|
| | k=1 | k=2 | k=4 | k=1 | k=2 | k=4 | k=1 | k=2 | k=4 | k=1 | k=2 | k=4 |
| | | | | | | VG-DENSE | | | | | | |
| GIN-70K | 0.162 | 0.199 | 0.268 | 0.659 | 0.669 | 0.695 | 0.162 | 0.244 | 0.380 | 0.201 | 0.257 | 0.340 |
| GAT-70K | 0.178 | 0.252 | 0.316 | 0.700 | 0.706 | 0.720 | 0.178 | 0.304 | 0.436 | 0.216 | 0.271 | 0.352 |
| GCN-70K | **0.248** | **0.295** | **0.372** | **0.742** | **0.734** | **0.747** | **0.248** | **0.364** | **0.490** | **0.280** | **0.330** | **0.405** |
| | | | | | | VG-RANDOM | | | | | | |
| GAT-70K | 0.184 | 0.244 | 0.288 | 0.696 | 0.696 | 0.713 | 0.184 | 0.292 | 0.380 | 0.112 | 0.174 | 0.266 |
| GIN-70K | 0.030 | 0.041 | 0.065 | 0.569 | 0.586 | 0.628 | 0.030 | 0.046 | 0.068 | 0.222 | 0.277 | 0.357 |
| GCN-70K | **0.214** | **0.249** | **0.300** | **0.697** | **0.701** | **0.715** | **0.214** | **0.300** | **0.418** | **0.250** | **0.302** | **0.380** |

## 4.3   EXTENDABILITY OF GRAPH-BASED COUNTERFACTUALS

The flexibility of our approach is proven under two separate scenarios: a) the application on unannotated images, b) the expansion into other modalities. For direct comparison to SC we provide global CEs by averaging overall graph triple edits (additions, deletions, substitutions).

**Unannotated datasets**   We replicate Dervakos et al. (2023)'s experiment on explaining the classification of web-crawled images into the categories 'driver' and 'pedestrian'. Here, the authors served as manual classifiers; thus, we explain a non-neural classifier (details in Appendix H). By employing the SOTA scene graph generator (SGG) of Cong et al. (2023) we extract global edits from generated graphs for the transition from 'pedestrian' to 'driver', as presented in Figure 5(left). Their relevance is verified by our common sense. For instance, people wear helmets when driving - addition of (helmet, on, head) and (man, on, bike) - and cover the bike seat with their body - deletion of (seat, on, bike). To validate our method's consistency across other annotation techniques, we replace the SGG with a

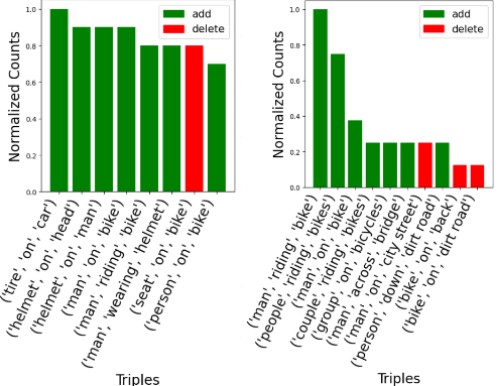

Figure 5: Graph edits (triples inserted/ deleted) to implement the 'pedestrian' → 'driver' transition.

captioning to graph-extraction-from-text pipeline, employing BLIP (Li et al., 2022) and Unified VSE (Wu et al., 2019) in that order. We confirm that resulting edits (Figure 5 (right)) semantically resemble

the ones produced using the SGG annotation process. The contribution of the graph structure is not lost in this experiment either. More accurate local edits are once again achieved through the consideration of the multiplicity of objects and relations.

**Audio classification**    While our primary focus is on images, we demonstrate our method's model-agnostic nature by applying it to other modalities, such as audio features following Dervakos et al. (2023). We provide CEs for the IEEE COVID-19 sensor informatics competition winner [1] trained on a subset of the Coswara Dataset, which predicts COVID-19 from cough audio. Our implementation relies on the Smarty4covid dataset Zarkogianni et al. (2023). Our analysis aligns with findings from the SC paper: It reveals that the most frequent alterations pertain to symptom-related concepts, particularly respiratory symptoms in Tab. 7, and uncovers the reported gender bias of the training dataset which includes more COVID-positive women than men. Note that presented edits are all additions and in the form ('User', 'symptom', X) where X is an element of Tab. 7. A longer list is in Appendix G.

Table 7: Global edits for COVID-19 Negative → Positive.

| Concepts | Norm. Counts |
|---|---|
| 'Sneezing' | 1.0 |
| 'RunnyNose' | 0.78 |
| 'DryThroat' | 0.35 |
| 'Fever' | 0.34 |
| 'Dizziness' | 0.31 |

### 4.4    Efficiency of graph-based counterfactuals

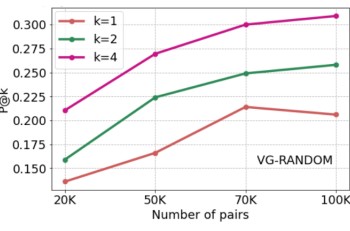

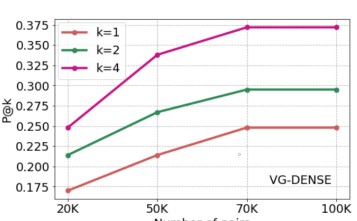

Figure 6: P@k of GCN variant for different training pairs $p$.

**Time Performance for Counterfactual Retrieval**    We experimentally confirm that our method allows for **efficient** counterfactual retrieval. In Tab. 8, we report execution times for counterfactual computation on the complete sets of graphs using GED (Fankhauser et al., 2011) versus our GNN-powered approach. We further report retrieval and inference time of our method. Even by adding times for all GCN-N/2 operations, we significantly relieve the computational burden of calculating the ground truth GED for all graph pairs, especially for larger graphs.

Table 8: Time (sec) for counterfactual calculation. Training time is reported due to the transductivity of the GNN method.

| | GED↓ | GCN-N/2 (train)↓ | GCN-N/2 (retr.)↓ | GCN-N/2 (infer.)↓ |
|---|---|---|---|---|
| CUB | 46220 | **32691** | 0.033 | 0.060 |
| VG-DENSE | 13982 | **12059** | 0.033 | 0.063 |
| VG-RANDOM | 18787 | **16271** | 0.029 | 0.099 |

**Performance-complexity trade-off**    In Fig. 6, we examine how precision varies using different numbers of training pairs $p$. The consistency of behavior exhibited over 70K pairs concludes our claim that **N/2 training pairs are adequate** for appropriate graph embedding using GCN.

## 5    Conclusion

In this paper, we proposed a new model-agnostic approach for counterfactual computation based on conceptual semantics and their respective relations. We leveraged the expressive power of scene graphs for image representation and suggested counterfactual retrieval by GED calculation. To this end, we used a GNN-based similarity model to accelerate the retrieval process, which would otherwise rely on solving an NP-hard problem for all input graph pairs. Comparison with previous counterfactual models proved that our explanations correspond to minimal edits and are more human interpretable, especially when interactions between concepts are dense, while still ensuring actionability. We further confirmed the applicability of our framework on image datasets with no annotations at all, utilizing scene generation techniques, as well as on audio data. As future work, we plan to explore potential limitations such as robustness and the impact of low quality annotations, in the setting of conceptual CEs, as well as further improve efficiency by employing unsupervised GNN methods for reduced training time, or by ensuring inductivity and in turn offline training.

---

[1]https://healthcaresummit.ieee.org/data-hackathon/ieee-covid-19-sensor-informatics-challenge/

## REPRODUCIBILITY STATEMENT

All the experiments detailed in this work are entirely reproducible. You can find an overview of our method in Section 3. The experimental configurations are outlined in Section 4, including further details in Section C of the Appendix, which also includes specific hyperparameters used for the GNN component. We have made the complete code available in a zip file, accompanied by a comprehensive README to guide its usage. Furthermore, additional data and results can be accessed through an anonymous URL, provided within the README.

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

# A    HUMAN EVALUATION DETAILS

## A.1    PARTICIPANTS AND CONSENT

We distributed an information sheet describing the goals and stages of our human surveys to software engineering students online. We clarified that their participation would be voluntary and without any form of compensation. We additionally distributed the following form to obtain annotators' consent in the form of a checklist. We used the same form both for the machine teaching as well as the counterfactual preference experiment. The 33 people who ultimately participated were young adults of ages 19-25 both male and female, without any knowledge of bird species.

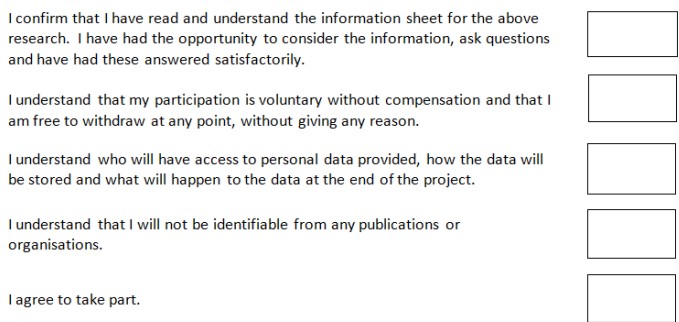

Figure 7: Screenshot of the consent form for human evaluation. Our annotators fill out this form before they proceed with annotations.

Our human survey was completely anonymous and we did not record any type of personal data from our annotators.

## A.2    1ST EXPERIMENT: COMPARATIVE HUMAN SURVEY

In Fig. 8, we present a screenshot of the platform we provided to our evaluators for the comparative user survey. Users are asked to select a sample to annotate, as shown in the panel of Fig. 9. We ensured that our evaluators can clearly view the images and their details by providing 'zoom-in'/'zoom-out' tools, as well as the ability to navigate within the image with the 'pan' and 'move' options.

An annotator can click on any sample to be annotated, thus moving to a screen such as the one of Figure 9. The source image is presented on the left, and the two alternative options (ours versus a counterfactual image of CVE (Vandenhende et al., 2022) or SC (Dervakos et al., 2023)) are placed in the middle and the rightmost column. These options are shuffled in each sample, so that no bias towards each choice is created. Only one of the options ("Image 1", "Image 2" or "Can't tell") can be selected for each sample.

In this first human experiment, our annotators can evaluate as many samples as they wish; however, they cannot update an existing annotation. All 33 annotators participated in this experiment.

## A.3    2ND EXPERIMENT: MACHINE-TEACHING HUMAN SURVEY

We once again employ the same platform as for the previous human experiment. However, this time each annotator can only evaluate **one** single sample; we enforce this restriction to clearly evaluate the contribution of the learning phase, excluding situations that an annotator could have become more 'competent' after passing many times through the learning phase.

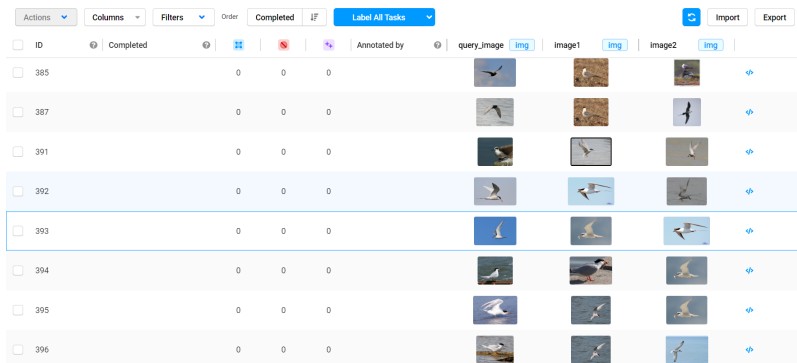

Figure 8: Screenshot of the platform provided for human evaluation.

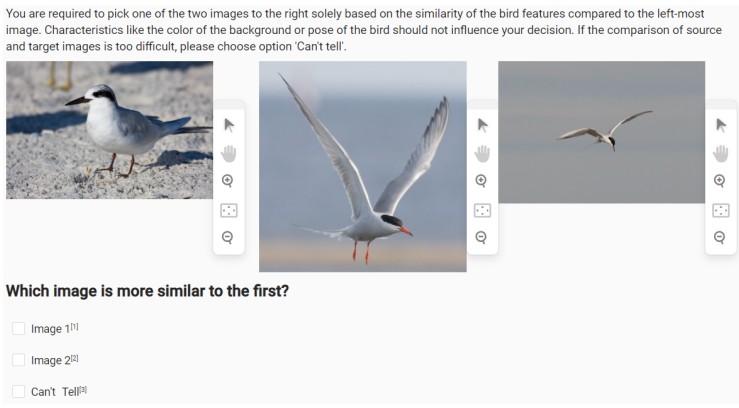

Figure 9: Annotation panel with instructions and image navigation tools provided to the evaluators for CUB.

The experimental workflow is adopted from Vandenhende et al. (2022), therefore we include all the three stages (pre-learning, learning and testing).

**Pre-learning stage**    In the pre-learning stage, users are presented with unlabeled images from the test set to get familiarized with the nature of the images they will be tasked to classify later on. Fig. 10 is provided as an example of the pre-learning screen. The annotators become aware that the classification to the anonymized classes A and B cannot be performed without passing through the learning stage, therefore selecting "I don't know" is the expected option. In Fig. 10, we can explicitly see the three options for image classification, namely "Class A", "Class B" or "I don't know". Only one can be selected at a time, as in Vandenhende et al. (2022).

**Learning stage**    The learning stage comprises the heart of this human experiment. As mentioned in the main paper, we perform two variants of it to measure the degree of reliance on concepts, according to human perception. A user can either participate in the "visually-informed" or the "blind" experiment, but not both. This is necessary so that we exclude the possibility of evaluating the same data sample in each of the experiments and thus eliminate the possibility of having some knowledge transfer across the two variants of this experiment. Annotators are divided into equal subgroups (17 in the "visually-informed" variant and 16 in the "blind" one).

In the **visually-informed** variant, annotators are presented with training images from anonymized classes A and B, together with their scene graphs, as shown in Figures 11, 12, 13. Of course, training and test images do not overlap. Annotators are again provided with 'zoom-in'/'zoom-out', 'pan', 'move' tools, etc. to navigate within the images and the accompanying scene graphs.

Training images on the left always belong to class A, while images on the right always belong to class B. Scene graphs on the right also contain the edits needed to perform the $A \rightarrow B$ transition,

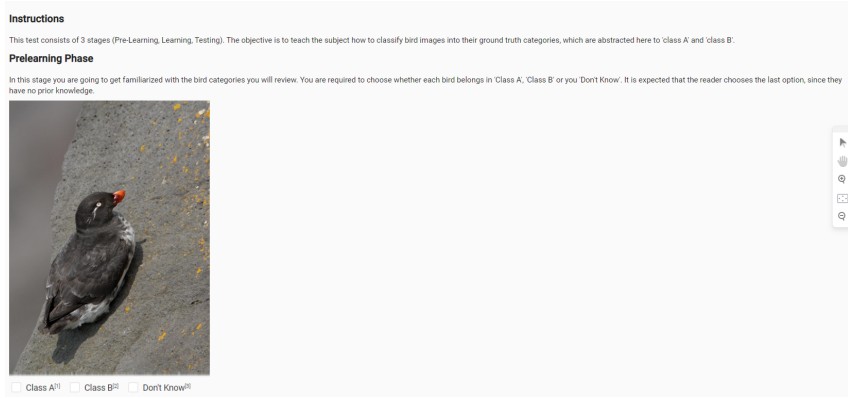

Figure 10: Pre-Learning stage instructions for CUB machine teaching experiment. Choices are "Class A", "Class B" or "I don't know".

with green nodes representing concept additions, blue nodes indicating concept substitutions (both source and target concepts of the substitution are shown), and red nodes denoting concept deletions. The rest of the nodes imply that the corresponding concepts remain the same between the two classes.

A user implicitly focuses on the most frequent insertions, substitutions, and deletions performed throughout the training stage to understand the discriminative features between class A and class B. Associating such concepts with the images helps mapping graph edits to visual differences so that the user learns to separate classes visually and conceptually.

In the **blind** variant of the learning stage, only scene graphs are provided, but no training images. Also, the graph edits are presented to the users via colored nodes. This learning variant is a direct analogy to the machine-teaching learning stage implemented by Vandenhende et al. (2022): in their case, pixels corresponding to discriminative regions that act as explanations are provided, while the rest of the bird image is blurred out. Therefore, annotators need to learn solely from the explanation and mentally connect the corresponding concepts to existing visual regions of the testing images. In our case, the derived explanations correspond to graph edits, therefore annotators have to learn the discriminative concepts that are added, substituted, or deleted to perform the $A \rightarrow B$ transition. However, since our learning setting is performed without any visual clue, we regard our blind learning stage as being **more difficult** than the learning stage that Vandenhende et al. (2022) implement; our annotators have to connect concepts with image regions, thus performing cross-modal grounding in order to learn discriminative features.

Throughout the blind learning stage, we are able to measure the reliance on concepts rather than pixels to learn to classify images of unknown classes. This experiment is important in order to highlight how meaningful and informative conceptual explanations are to humans, so that they can approximate a zero-shot classification setting.

**Testing stage**    In the testing stage, users are provided with the same images as in the pre-learning stage. No scene graphs are provided. Based on the previous stage, annotators should have learned visual and conceptual differences between classes; therefore, they are tasked to assign an appropriate class to each test image, by selecting either "class A" or "class B" for each of them. Contrary to the pre-learning stage, the option "I don't know" is not provided.

After this stage, an accuracy score is extracted per user, based on their correct selections in the testing stage. We then extract an average accuracy per user, which we report in Tab. 4 of the main paper. Our average accuracy for the visually-informed experiment is 93.88%, indicating that in most cases users are highly capable of recognizing the key concepts that separate the two given bird classes, grounding them with visual information. As for the blind experiment, the average testing accuracy is 89.28%. Being rather close to the visually-informed accuracy percentage, we can safely assume that **concepts are more than adequate** towards teaching discriminative characteristics to humans, even if they lack association with purely visual information. Both visually-informed and blind accuracy scores clearly

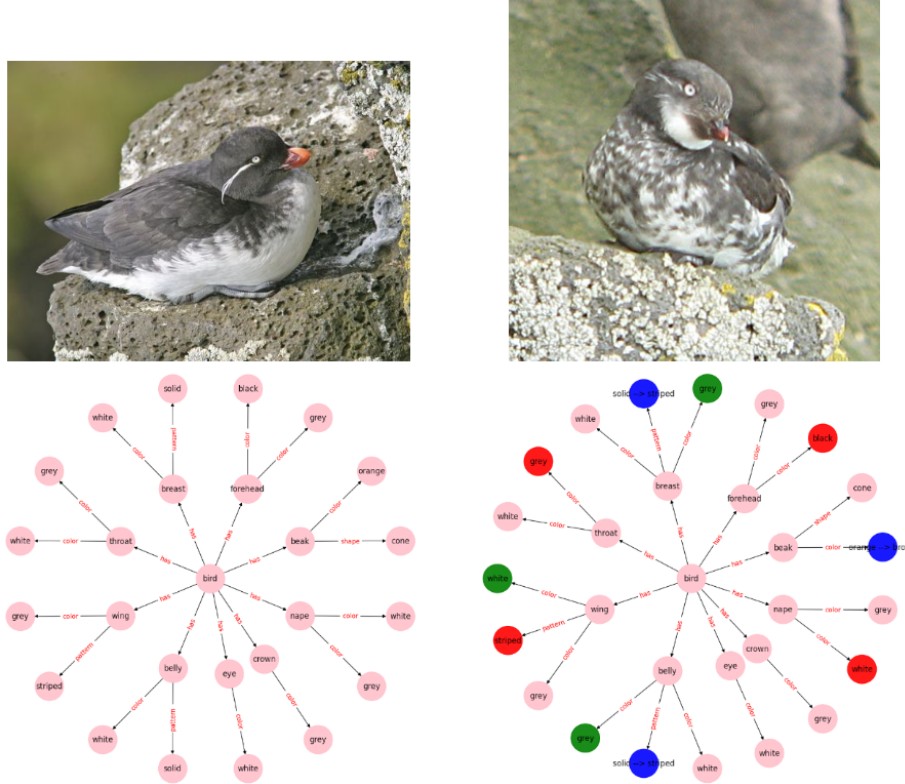

Figure 11: Example of the visually-informed learning stage.

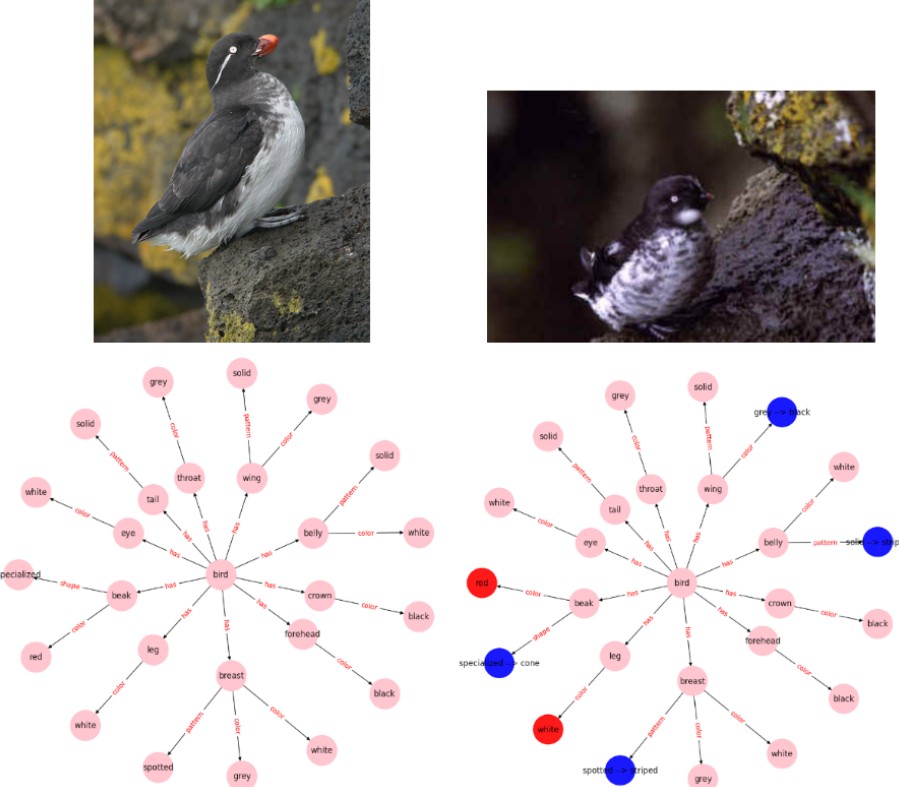

Figure 12: Example of the visually-informed learning stage.

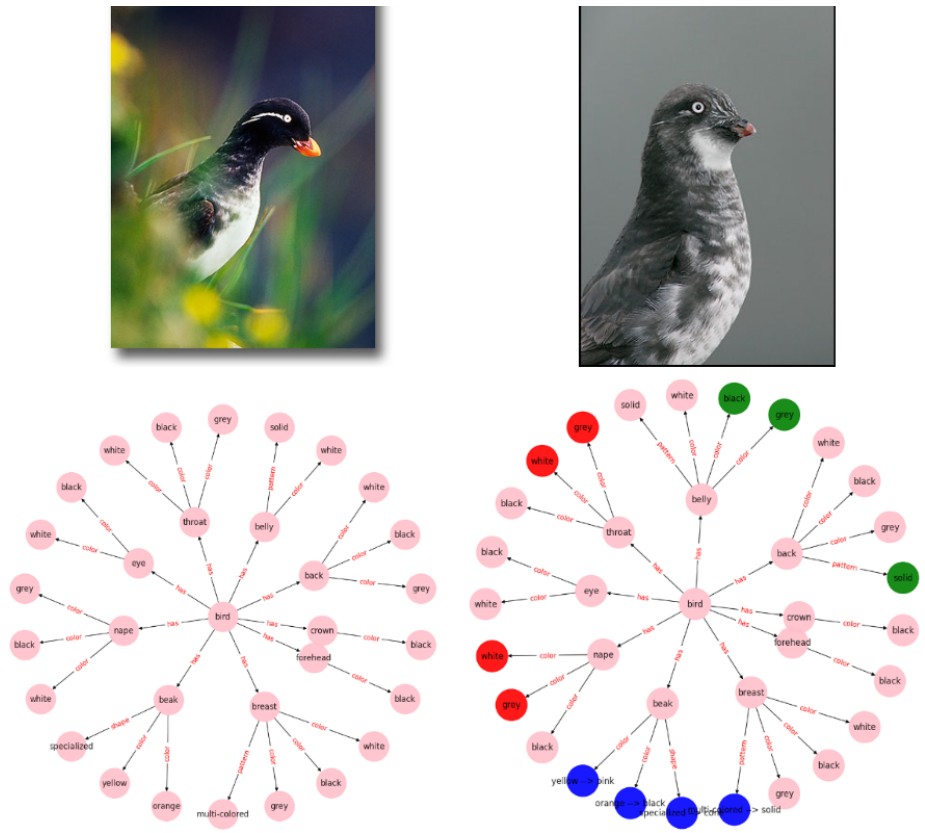

Figure 13: Example of the visually-informed learning stage.

outperform the accuracy scores reported in CVE, demonstrating that conceptual explanations are more **meaningful** and **informative** to humans compared to pixel-level explanations.

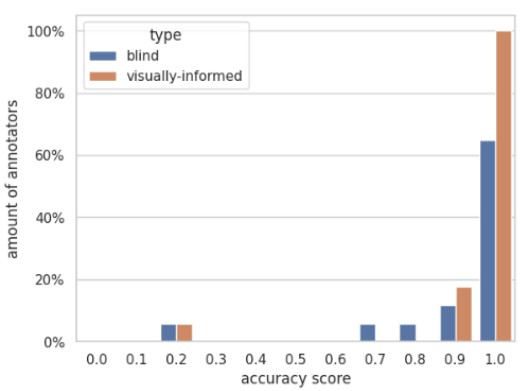

Figure 14: Distribution of test accuracy for machine teaching human evaluation experiments.

**Accuracy score distribution** In Fig. 14, we present a more detailed analysis of the accuracy scores achieved by human subjects during the testing phase of the machine teaching experiment. It is apparent that scores peak at 0.9 and 1.0; thus, explanations produced by our method are highly human-interpretable and beneficial to perform classification. Comparison between 'visually-informed' and 'blind' results reveals that the decrease in test accuracy for the experiment without a visual aid is gradual.

**Applicability of machine-teaching experiment** The machine-teaching experiment is purposely run exclusively on the CUB dataset. To highlight the merits of the learning phase: annotators have no knowledge of bird species, therefore they can highly benefit from learning discriminative bird attributes, and then apply this new knowledge in the testing phase. For example, none of the annotators knows the difference between a Parakeet Auklet and a Least Auklet. Nevertheless, after the learning stage, they are able to recognize the basic discriminative attributes, which will help them classify instances of the test phase. On the other hand, Visual Genome contains images of common everyday scenes, rendering a similar experiment rather redundant in such instances. For example, a human already knows key concepts

that discriminate a kitchen from a bedroom, therefore the learning stage would be of no value, even if the scene labels are anonymized. We can view this scenario as an analog to data leakage.

Moreover, there is always a possibility that some concepts can be misleading. In such cases, we expect visual classifiers to present a bias towards such concepts, while this is not the case for humans. For example, a TV can be present in both kitchens and bedrooms. However, in a hypothetical scenario that selected bedroom images have TVs, but kitchen images do not, the graphs that serve as explanations would contain many "add TV" nodes. Therefore, a human expects to classify images containing TVs in one class, and images that do not contain TVs in the other (as a visual classifier would do if trained on such data). But when finally humans are presented with real test images, they will not be misled by the presence or the absence of TVs, but rather rely on their commonsense to perform classification. Thus, not only is the learning stage redundant, but the obvious existing bias "add TV" is not reflected in the final classification; in this case, the counterfactual explanation itself would be of no value to humans.

## B  GRAPH STATISTICS

In Table 9 we present some statistics regarding the graphs of the datasets used in our work. VG-DENSE and VG-RANDOM contain 500 graphs each, CUB contains 422 graphs, D/P-SGG and D/P-CAPTION denote the web-crawled datasets of Section 4.3 with 259 graphs each and SMARTY denotes the COVID-19 classification dataset with 548 graphs. Table 10 contains additional statistics about datasets utilized only in the appendix. These are GQA (Hudson & Manning, 2019) with 500 graphs mentioned in Sec. D.3 and Action Genome (AG) (Ji et al., 2020) with 300 graphs mention in Sec. F. The size and density of input data should be considered when viewing results in the experimental section.

Table 9: Statistics regarding graphs of different datasets used in the main paper.

|  |  | VG-DENSE | VG-RANDOM | CUB | D/P-SGG | D/P-CAPTION | SMARTY |
|---|---|---|---|---|---|---|---|
| Mean | density | 0.199 | 0.061 | 0.037 | 0.128 | 0.247 | 0.234 |
|  | edges | 9.042 | 8.768 | 27.519 | 9.367 | 1.756 | 4.398 |
|  | nodes | 7.252 | 14.566 | 28.519 | 9.733 | 3.197 | 5.398 |
|  | isolated nodes | 0.47 | 3.37 | 0 | 0.323 | 0.901 | 0 |
| Max | density | 0.467 | 0.667 | 0.111 | 1.0 | 0.5 | 0.333 |
|  | edges | 36 | 27 | 53 | 18 | 4 | 15 |
|  | nodes | 15 | 20 | 54 | 18 | 5 | 16 |
|  | isolated nodes | 3 | 12 | 0 | 3 | 4 | 0 |
| Min | density | 0.144 | 0.013 | 0.019 | 0.046 | 0.05 | 0.062 |
|  | edges | 5 | 5 | 8 | 1 | 1 | 2 |
|  | nodes | 6 | 4 | 9 | 2 | 2 | 3 |
|  | isolated nodes | 0 | 0 | 0 | 0 | 0 | 0 |

## C  EXPERIMENTAL SETTINGS

In addition to details regarding resources used for the experimental setup mentioned in the main paper, we further report specific training configurations for GNN models. All presented results were achieved using single-layer GNNs of a dimension of 2048, built as explained in Sec. 3 of the main paper. For reproducibility purposes, we report that these models were optimized for a batch size of 32 and trained for 50 epochs, without the use of dropout. The employed optimizer was Adam without weight decay. The respective learning rate varied among GNN variants. To be precise, we used a learning rate of 0.04 for GCN and 0.02 for GAT and GIN. GAT and GIN also have model-specific hyperparameters - attention heads and the learnable parameter epsilon respectively. Best results were achieved by leveraging 8 attention heads and setting epsilon to non-learnable.

Last but not least, an important hyperparameter of the GNN models is the number of training pairs, denoted as $p$. As explained, optimal models used $p \approx N/2$, which varies among datasets. Specifically,

Table 10: Statistics regarding graphs of different datasets in the appendix.

|      |                | AG     | GQA   |
|------|----------------|--------|-------|
| Mean | density        | 0.189  | 0.241 |
|      | edges          | 13.227 | 8.144 |
|      | nodes          | 8.84   | 6.66  |
|      | isolated nodes | 1.173  | 1.366 |
| Max  | density        | 0.45   | 1.0   |
|      | edges          | 51     | 20    |
|      | nodes          | 17     | 12    |
|      | isolated nodes | 2      | 15    |
| Min  | density        | 0.1    | 0.125 |
|      | edges          | 4      | 5     |
|      | nodes          | 5      | 4     |
|      | isolated nodes | 0      | 0     |

the parameter $p$ is set to 70K for datasets with 500 graphs, 50K for datasets with 422 graphs, and 25K for datasets with 300 graphs. However, we also conducted ablations on the number of training pairs, setting $p$ to values reported in Fig. 5 of the main paper. In those cases, we explored using 16%, 40%, and 80% of the existent graph pairs, in addition to the "golden" 50%.

Regarding graph kernels that were employed for comparison, we report that the Pyramid Match kernel was used with its default settings. The settings include leveraging labels, a histogram level $L$ of 4, and hypercube dimensions $d$ of 6.

The classifier in our CUB experiments (ResNet-50) was chosen in alignment to experiments performed in the works compared and recreated here. As for the choice of the Places365 instead of a pretrained ImageNet classifier, it was conscious. Despite the latte being potentially more widely recognized and researched, it is trained on the ImageNet dataset, which primarily consists of foreground objects. In Visual Genome, the majority of instances depict scenes, providing substantial background. Although some instances focus more on specific objects, they are still situated within a particular environment. In contrast, ImageNet classifiers face challenges with such inputs, as only about 3% of the target classes in the corresponding dataset pertain to broader scenes. Classifiers for the rest of the datasets are explained in detail in the following sections.

The code for all experiments is provided within the zip file of the supplementary material, accompanied by comprehensive instructions.

## D QUANTITATIVE EXPERIMENTS

### D.1 GRAPH KERNELS

Graph kernels are kernel functions used on graphs that measure similarity in polynomial time, providing an efficient and widely applicable alternative to GED. In the context of this paper, we experimented with several kernels from the GraKeL library (Siglidis et al., 2020), as a baseline measure for counterfactual retrieval. Our goal is to guarantee that our GNN framework outperforms such methods. We present results from the best-performing kernel Pyramid Match.

**Pyramid Match (PM) kernel** The PM (Grauman & Darrell, 2007; Nikolentzos et al., 2017) graph kernel operates by initially embedding each graph's nodes in a d-dimensional vector space using the absolute eigenvectors of the largest eigenvalues of the adjacency matrix. The sets of graph vertices are compared by mapping the corresponding points in the d-dimensional hypercube to multi-resolution histograms, using a weighted histogram intersection function. The comparison process occurs in several levels, corresponding to different regions of the feature space with increasing size. The algorithm counts new matches at each level - i. e. points in the same region - and weights them according to the size of the level. The cells/regions double in size in each iteration of the algorithm.

This procedure is applicable to graphs with node/edge labels; thus, we cannot leverage GloVe embeddings for initialization. Matches exist only between points with the exact same label. The overall complexity of the algorithm is $O(ndL)$, which compared to other kernel methods is quite computationally expensive.

**Ranking results for VG** using the PM kernel are presented in Tab. 11 (as an extension of Tab. 6 presented in the main paper). It is obvious that PM-kernel lacks the representational capacity offered by GNNs, especially in comparison to our best performer, GCN-70K.

Table 11: Ranking results on the two VG variants for graph kernels vs our best performing GNN.

| Models | P@k ↑ | | | NDCG@k ↑ | | | P@k (binary) ↑ | | | NDCG@k (binary) ↑ | | |
|---|---|---|---|---|---|---|---|---|---|---|---|---|
| | k=1 | k=2 | k=4 | k=1 | k=2 | k=4 | k=1 | k=2 | k=4 | k=1 | k=2 | k=4 |
| | VG-DENSE | | | | | | | | | | | |
| PM-kernel | 0.132 | 0.134 | 0.174 | 0.651 | 0.657 | 0.678 | 0.132 | 0.186 | 0.258 | 0.191 | 0.248 | 0.331 |
| GCN-70K | **0.248** | **0.295** | **0.372** | **0.742** | **0.734** | **0.747** | **0.248** | **0.364** | **0.490** | **0.280** | **0.330** | **0.405** |
| | VG-RANDOM | | | | | | | | | | | |
| PM-kernel | 0.000 | 0.003 | 0.011 | 0.547 | 0.571 | 0.620 | 0.000 | 0.002 | 0.014 | 0.098 | 0.161 | 0.254 |
| GCN-70K | **0.214** | **0.249** | **0.300** | **0.697** | **0.701** | **0.715** | **0.214** | **0.300** | **0.418** | **0.250** | **0.302** | **0.380** |

## D.2 AVERAGE GED

In addition to the average number of edits metric and the ranking metrics using the ground truth GED as the golden standard, we present the average GED of the top-1 counterfactual results. This supplementary measure serves to explicitly enhance comprehension of the significance of semantic context. Notably, within the main paper, our qualitative results illustrate scenarios where, despite an equal (or lower) number of edits, the GED can at times be higher. This divergence arises because edits are not uniformly weighted but rather based on their semantic similarity.

Table 12: Average top-1 GED on CUB.

| | CUB |
|---|---|
| CVE | 257.189 |
| SC | 263.795 |
| Ours | **211.687** |

Table 13: Average top-1 GED on VG.

| | VG-DENSE | | VG-RANDOM | |
|---|---|---|---|---|
| | Normal | Refined | Normal | Refined |
| SC | 105.92 | 128.669 | 161.368 | 186.770 |
| Ours | **104.368** | **122.411** | **159.722** | **180.674** |

Table 14: Refined average number of node, edge & total edits on VG.

| | VG-DENSE | | | VG-RANDOM | | |
|---|---|---|---|---|---|---|
| | Node↓ | Edge↓ | Total↓ | Node↓ | Edge↓ | Total↓ |
| SC | **4.73** | 7.65 | 12.38 | **11.96** | **7.48** | **19.44** |
| Ours | 5.07 | **6.96** | **12.03** | 12.37 | 7.52 | 19.89 |

For the VG dataset, we present results comparing "Normal" and "Refined" outcomes, as shown in Table 13. In this context, "Refined" denotes presenting averages exclusively when the two methods yield distinct counterfactuals. We adopted this approach due to the observation that 75% of CEs for VG-DENSE and 73% for VG-RANDOM were identical between methods, creating an impression of increased result proximity. To provide a comprehensive view, we also furnish more refined average

number of edits results in Table 14. Notably, for the CUB dataset, such an analysis is unnecessary; nonetheless, we include the average top-1 GED in Table 12.

### D.3 ADDITIONAL DATASETS

**Ranking results for GQA**  The analysis performed on Visual Genome (VG) is extended on the GQA dataset (Hudson & Manning, 2019). In fact, GQA comprises a variant of VG focusing on compositional question-answering involving real-world scenes. Since GQA images and accompanying scene graphs are very similar to the ones involved in our VG analysis, the obtained results verify the findings reported for VG without offering other novel insights. In Tab. 15 we present per-model results for 70K training pairs. GCN remains the most powerful architecture compared to the other ones, an observation validating the findings reported for the rest of the datasets.

Table 15: Ranking results on GQA for different graph models.

|  | P@k ↑ | | | NDCG@k ↑ | | | P@k (binary) ↑ | | | NDCG@k (binary) ↑ | | |
| --- | --- | --- | --- | --- | --- | --- | --- | --- | --- | --- | --- | --- |
|  | k=1 | k=2 | k=4 | k=1 | k=2 | k=4 | k=1 | k=2 | k=4 | k=1 | k=2 | k=4 |
| PM | 0.060 | 0.104 | 0.046 | 0.652 | 0.653 | 0.676 | 0.104 | 0.146 | 0.202 | 0.165 | 0.223 | 0.310 |
| GIN-70K | 0.156 | 0.244 | 0.285 | 0.697 | 0.699 | 0.721 | 0.156 | 0.274 | 0.392 | 0.195 | 0.252 | 0.335 |
| GAT-70K | 0.126 | 0.172 | 0.222 | 0.663 | 0.675 | 0.693 | 0.126 | 0.214 | 0.302 | 0.177 | 0.235 | 0.320 |
| GCN-70K | **0.188** | **0.286** | **0.341** | **0.725** | **0.730** | **0.740** | **0.188** | **0.332** | **0.478** | **0.221** | **0.276** | **0.356** |

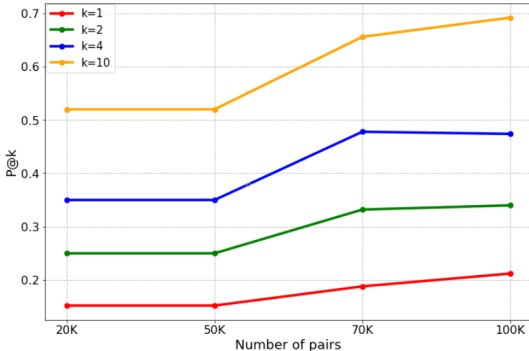

Figure 15: Comparison of the GCN performance measured in P@k for different number of training pairs $p$ for GQA.

**Performance-complexity trade-off for GQA** In Fig. 15 we present the performance analysis for different numbers of training pairs $p$ on the GQA dataset, focusing on our best-performing model (GCN). Once again, $N/2 \sim 70K$ pairs are adequate for learning proper representations of scene graphs, validating our initial claim that GED does not have to be computed for more than $N/2$ graph pairs to obtain a satisfactory approximation.

**Ranking results for Action Genome** are presented in Tab. 16, while **number of edits for Action Genome** is presented in Tab. 17, both for $N/2 = 25K$.

Table 16: Ranking results on AG.

|  | P@k ↑ | | | NDCG@k ↑ | | | P@k (binary) ↑ | | | NDCG@k (binary) ↑ | | |
| --- | --- | --- | --- | --- | --- | --- | --- | --- | --- | --- | --- | --- |
|  | k=1 | k=2 | k=4 | k=1 | k=2 | k=4 | k=1 | k=2 | k=4 | k=1 | k=2 | k=4 |
| GCN-25K | 0.167 | 0.212 | 0.266 | 0.695 | 0.702 | 0.718 | 0.167 | 0.26 | 0.41 | 0.211 | 0.266 | 0.348 |

Table 17: Average number of node, edge & total edits on AG.

|  | Node↓ | Edge↓ | Total↓ |
| --- | --- | --- | --- |
| GCN-25K | 4.87 | 7.99 | 12.86 |

### D.4 GLOBAL EDITS ON CUB

By aggregating edits from each image participating in the dataset, we can extract *global edits*: they describe what needs to be changed in total to explain the transition from one class to the other. These edits are more meaningful in the form of graph triples, but we can also provide concept or relationship edits. In Figure 16a, we provide the triple edits to explain the Parakeet Auklet → Least Auklet

counterfactual transition. Similarly, in Figure 16b we present global edits for concepts appearing on CUB images. The results align with human perception.

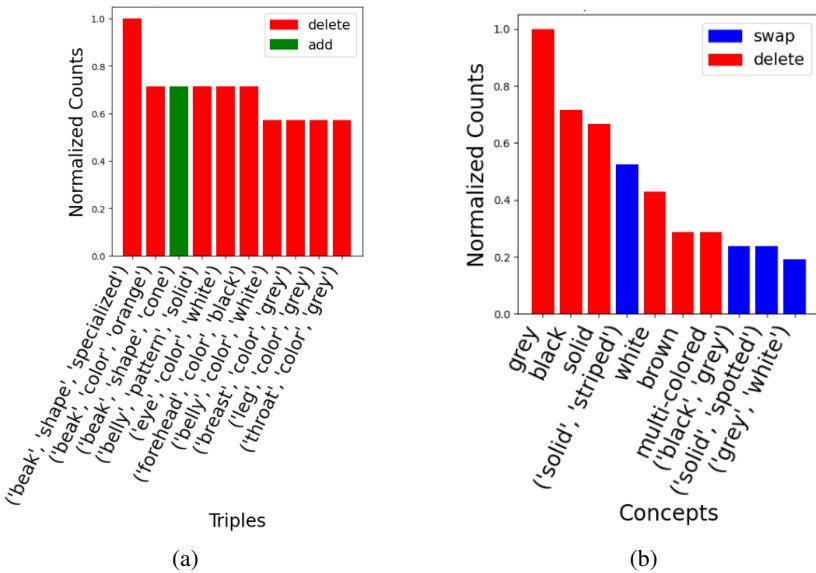

(a)          (b)

Figure 16: Triple and concept edits (insertions, deletions, substitutions) to perform Parakeet Auklet → Least Auklet transition.

# E    QUALITATIVE ANALYSIS

## E.1    COUNTERFACTUAL GRAPH GEOMETRY ON CUB

Our framework is capable of retrieving counterfactual graphs that not only respect node and edge semantics, but also graph geometry. This observation corresponds to more accurate retrieval capabilities that focus on semantic information regarding bird species without being significantly distracted from irrelevant characteristics such as the background. This can be an encouraging characteristic of our counterfactuals towards more robust explanations, even though this aspect is not analyzed in the current paper. First, we present a qualitative example of this claim. In Figure 17, we search for the most similar image to 17a using the method of CVE and ours.

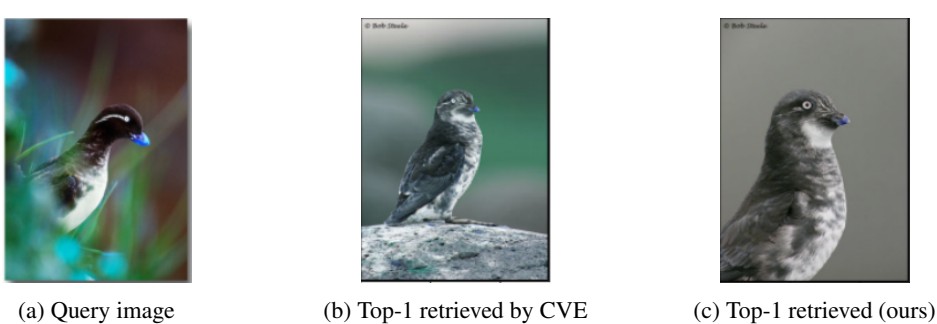

(a) Query image      (b) Top-1 retrieved by CVE      (c) Top-1 retrieved (ours)

Figure 17: A counterfactual explanation example.

Apparently, both counterfactual images are visually similar, as appearing in Fig. 17b and 17c. However, the representation power of scene graphs becomes evident in this case. In Fig. 18 we present the scene graphs corresponding one-to-one to the images of Fig. 17. The most similar graphs of 18a correspond to the graph of 18b according to CVE and 18c according to our approach. It is evident that our approach can successfully retrieve graphs that **better respect the geometry** of the source image scene graph. Another observation is that our approach manages to retrieve an

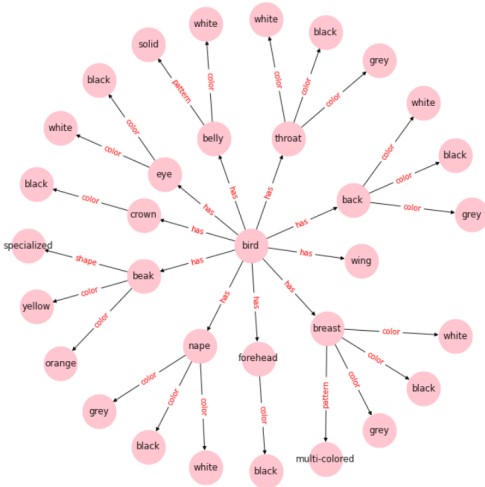

(a) Graph of class Parakeet Auklet corresponding to query image of Fig. 17a.

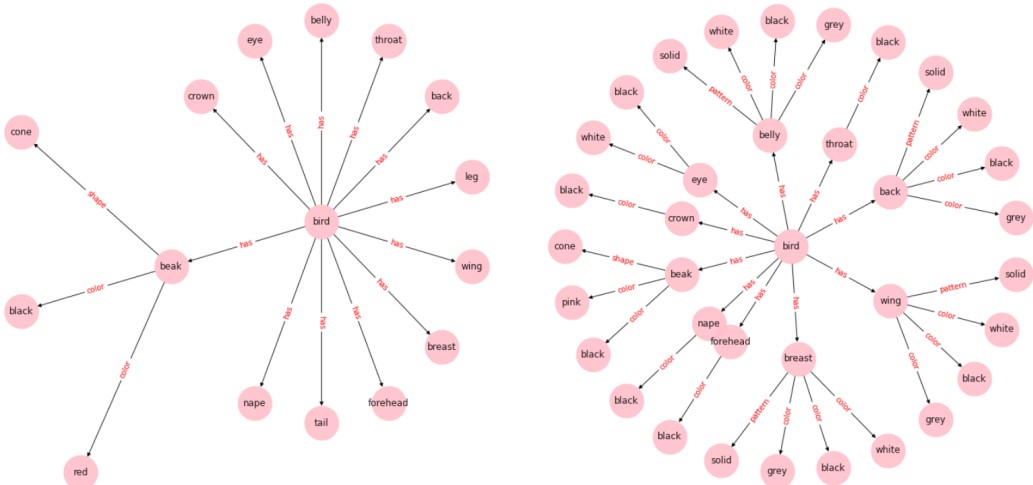

(b) Counterfactual graph of target class Least Auklet corresponding to Fig. 17b (as retrieved by CVE).

(c) Counterfactual graph of target class Least Auklet for Fig. 17c (as retrieved by our GCN-70K).

Figure 18: Example of scene graph structures of counterfactual graphs for Parakeet Auklet → Least Auklet class transition.

image without the concepts 'leg' or 'tail' which is more accurate compared to the source. Therefore, structural similarity leads to better semantic consistency.

### E.2 GRAPHS OF VISUAL GENOME

In Fig. 19 (VG-DENSE) and 20 (VG-RANDOM) we present the corresponding graphs to counterfactual images of Visual Genome produced by our method and the method of SC Dervakos et al. (2023), as presented in Fig. 4 of the main paper.

Inspection of VG-DENSE graphs clearly indicates that our method retrieves counterfactual instances that not only have similar concepts on nodes and edges but are also **structurally closer**. Suggesting counterfactual images with emphasis on object interactions leads to more accurate and meaningful explanations. For instance, in the first column, the relation 'surfer riding board' translates to 'man on board' for our method, whereas for SC (Dervakos et al., 2023) the man is essentially holding the board ('cord on board', 'cord on man').

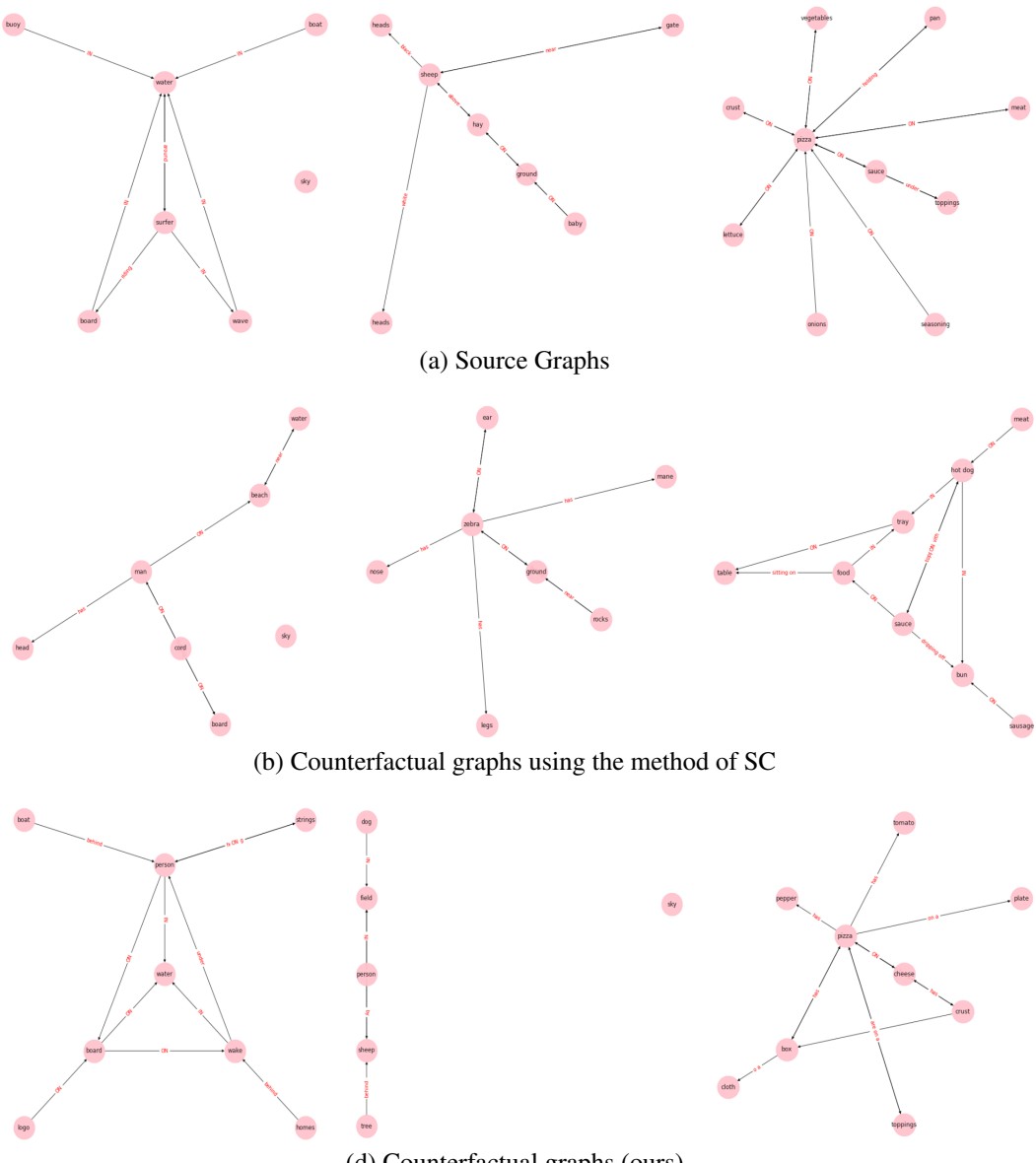

Figure 19: Qualitative Results on graphs for counterfactuals presented in Fig. 4 of the main paper for VG-DENSE.

In the case of VG-RANDOM where graphs have many isolated nodes and fewer edges, the comparison is not as straightforward. In columns 1 and 2 of Fig. 20, our method retrieves visually more similar instances by combining semantics and structure; thus, managing to preserve the main interacting concept of the image. However, when relations are sparse in the source graph, a greater amount of similar concepts will lead to better counterfactuals.

### E.3 ACTIONABILITY OF EDITS

We present two non-actionable counterfactual explanations produced by CVE and leverage their method of converting visual CEs into natural language. This approach enables us to precisely define changes between the query and counterfactual instances, which would be challenging with purely visual information. Having emphasized the importance of high-level semantics for human-

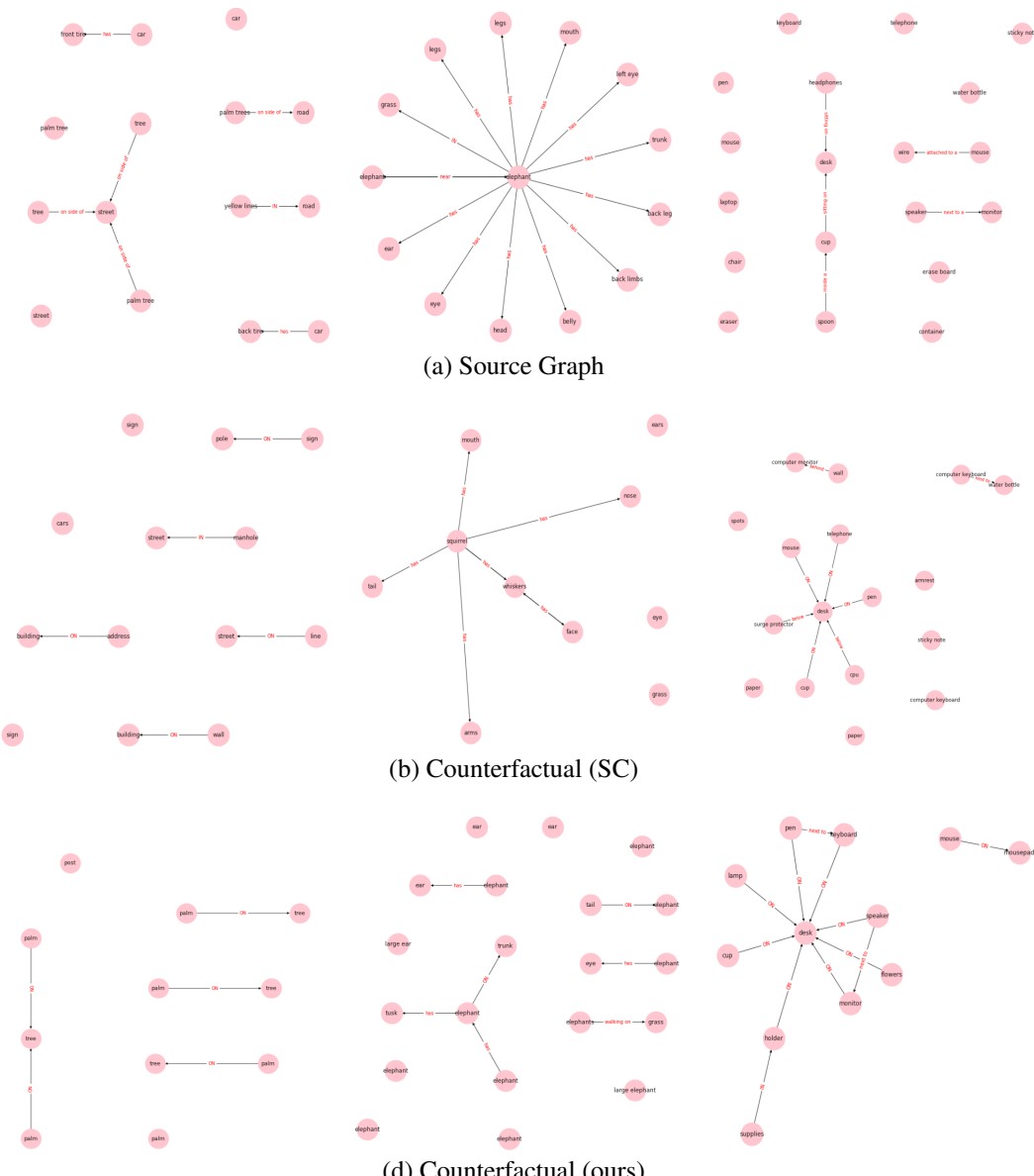

(a) Source Graph

(b) Counterfactual (SC)

(d) Counterfactual (ours)

Figure 20: Qualitative Results on graphs for counterfactuals presented in Fig. 4 of the main paper for VG-RANDOM.

interpretable CEs, we evaluate the inferred explanations based on linguistic cues rather than pixel-level edits. Both provided examples are deemed successful explanations.

Vandenhende et al. (2022) often propose single edits on the query image (left images of Figs. 21a, 21b) and deem them sufficient for the transition from query to target class. However, as explained in Sec. 4.1 of the main paper, this approach disregards the rest of the edits needed to be made between $I_{(A)}$ and $I_{(B)}$ and leads to instances that are in fact out-of-distribution. In the main paper, we gave an example that corresponds to Fig. 21a. In addition to the combination ('has_head_pattern::eyering', 'has_breast_color::grey') that was reported in text, we provide several other attribute combinations that do not exist in any other bird of the target class in Tab. 18. Furthermore, we present one more example in Fig. 21b. Vandenhende et al. (2022) claim that removal of the brown color from the crown of the Black billed cuckoo in Fig. 21b (left) is sufficient for it to be classified as a Yellow billed cuckoo.

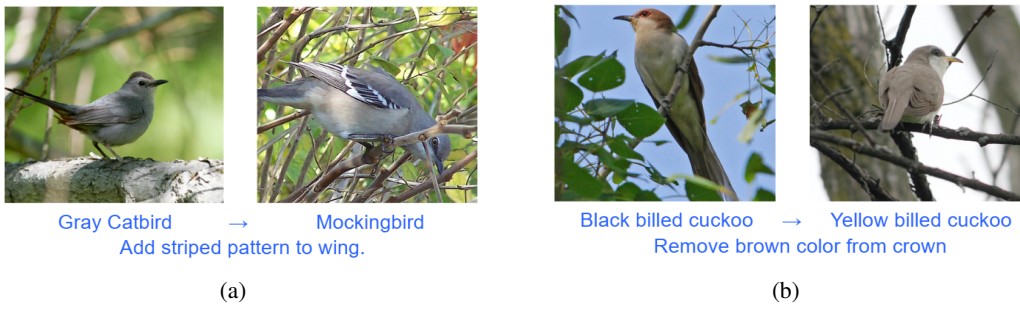



Gray Catbird     →     Mockingbird        Black billed cuckoo    →    Yellow billed cuckoo
Add striped pattern to wing.            Remove brown color from crown

(a)                                (b)



Figure 21: Counterfactual images from CVE and the proposed explanations using natural language.

After performing such an edit we obtain a new bird instance that retains the same features as the bird depicted in Fig. 21b (left), except it no longer has a brown crown. By generating all pairs of attributes of this new bird, we discover that none of the attribute pairs listed in Tab. 18 are representative of any bird in the target class (Yellow billed cuckoo).

It is straightforward to understand that more examples can easily be found throughout the dataset. Given the definition of target classes used in this example (most frequently confused by the classifier), counterfactual pairs are generally visually and semantically close. If we chose a different definition of the target class and picked one that is dissimilar to the query class, we can deduce that the list of out-of-distribution attribute combinations would be much longer.

Table 18: Out of distribution attribute pairs for target classes.

| Gray Catbird → Mockingbird |
|---|
| ('has_head_pattern::eyering', 'has_breast_color::grey') |
| ('has_head_pattern::eyering', 'has_belly_color::grey') |
| ('has_breast_color::grey', 'has_nape_color::brown') |
| ('has_breast_color::grey', 'has_shape::swallow-like') |
| ('has_upper_tail_color::white', 'has_wing_shape::pointed-wings') |
| ('has_breast_color::grey', 'has_primary_color::brown') |
| ('has_throat_color::grey', 'has_shape::swallow-like') |
| ('has_belly_color::grey', 'has_shape::swallow-like') |
| ('has_shape::swallow-like', 'has_leg_color::black') |
| **Black billed → Yellow billed Cuckoo** |
| ('has_upperparts_color::buff', 'has_upper_tail_color::white') |
| ('has_back_color::white', 'has_head_pattern::plain') |
| ('has_upper_tail_color::white', 'has_head_pattern::plain') |
| ('has_upper_tail_color::white', 'has_size::very_small_(3_-_5_in)') |
| ('has_upper_tail_color::white', 'has_back_pattern::solid') |
| ('has_upper_tail_color::white', 'has_leg_color::buff') |
| ('has_head_pattern::plain', 'has_nape_color::white') |
| ('has_nape_color::white', 'has_back_pattern::solid') |
| ('has_nape_color::white', 'has_tail_pattern::solid') |
| ('has_size::very_small_(3_-_5_in)', 'has_bill_color::grey') |
| ('has_leg_color::buff', 'has_bill_color::grey') |

Regarding our method, actionability, in the sense of counterfactuals being representative of the data distribution, is inherent. This guarantee arises from the fact that counterfactuals are actual samples from the target class, specifically the most similar ones to the query, and that we offer complete explanations. To be precise, the proposed counterfactual explanations consist of lists of all graph edits needed to transit from query $I_{(A)}$ to target $I_{(B)}$.

### E.4 ADDITIONAL RESULTS

**CUB** In Fig. 22 we provide some additional visual results of counterfactuals comparing our method with SC and CVE. Despite the visual similarity of the retrieved counterfactual images given all three methods, our approach consistently achieves significantly fewer number of total edits.

## F APPLICABILITY TO UNANNOTATED DATASETS

Applicability to unannotated datasets is a valid concern given our approach's dependence on scene graphs.

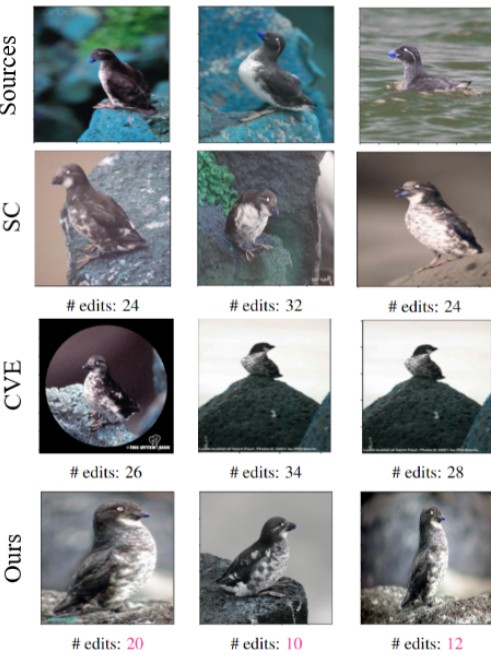

As previously established, graphs of images can be obtained either through manual annotations or automated construction methods. However, not all datasets have such readily available resources, therefore we invest our efforts around proving the applicability of our proposed approach to completely unannotated datasets.

Studying the impact of annotations is an important aspect, since an intrinsic characteristic of semantic explanations is their dependence on the knowledge of the individual that provides them. This inherent explainability attribute impacts systems in the same way it does humans. The knowledge supplied to an explainer will determine the specificity and scope of the explanations. Selecting the appropriate annotation technique is a critical step in receiving the desired breadth and depth of explanations.

In the following experiments, we explore these concerns by extracting counterfactual explanations via our proposed framework on unannotated datasets. Our framework is able to explain *any* classifier in a black-box manner, either being a non-neural classifier (humans in the case of the pedestrian vs driver experiment) or a convolution-based model (Zhou et al., 2017) (in the case of Action Genome).

Figure 22: Additional qualitative results of counterfactuals of the source class Parakeet Auklet belonging to target class Least Auklet. We also provide number of total edits per method, with colored instances denoting best results.

**Web images: pedestrian vs driver** Dervakos et al. (2023) gather images from Google, Bing, and Yahoo search engines corresponding to 'people', 'motorbikes', and 'bicycles' keywords and their combinations, and then manually split them in 'pedestrian' and 'driver' classes. Finally, 190 'driver' images were obtained (63 images of bicycle drivers and 127 of motorcycle drivers) and 69 'pedestrian' images (31 images of people and parked bicycles, and 38 images of people and parked motorcycles). Those classes are also adopted by us to highlight the importance of relationships (as claimed in Dervakos et al. (2023)), as well as extend this claim to support the usage of graphs over the relationship roll-up of SC. By rolling up the roles and converting them into concepts, we might unintentionally overlook important details for a given task. For example, when examining an image depicting a person on a motorbike in a store, alongside another motorbike on the street, by inspecting the scene graph, it is easy to assume that the scene represents a dealership, with the person testing the motorbike for potential purchase, without actually driving it. However, as Dervakos et al. (2023) encode this information with the objects: $person, riding\hat{\ }motorbike, motorbike, in\hat{\ }store$, and $motorbike, on\hat{\ }road$, they lose the distinction of which motorbike the user is actually riding, potentially leading to erroneous explanations. Nevertheless, leveraging the information within the graph allows us to arrive at more accurate conclusions, especially in fields as critical as Explainable Artificial Intelligence (XAI).

Apart from providing triple edits to explain the 'pedestrian' vs 'driver' classification (Figure 5), we also provide global relationship edits to discover if they are meaningful on their own. Indeed, relationship edits are meaningful in general, especially since the 'riding' relationship is inserted frequently (Figure 23, left plot corresponds to immediately deriving the SGG from the image, while the plot on the left denotes the edits occurring from captioning and then obtaining the graph from the caption). Moreover, the relationship 'on' appears frequently (in the SGG case), again confirming the action of sitting on a bike/motorcycle in order to drive.

Similarly, we extract global edits for concepts discriminating the pedestrian/driver categories. These edits are presented in Figure 24. By observing these plots (SGG - left, captioning and graph from text - right) we conclude that these edits are not really meaningful according to human perception: inserting wheels does not explain the 'pedestrian' → 'driver' transition, since in both classes bike/motorbike

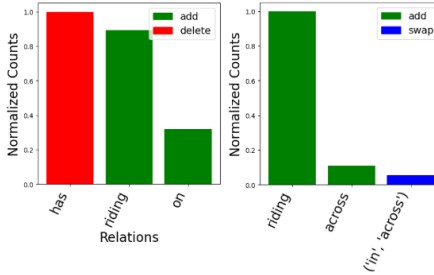

Figure 23: Relationships inserted/ deleted/substituted to implement the 'pedestrian' → 'driver' transition.

wheels may appear as part of these vehicles. The same observation is valid for the rest of the concepts appearing on these plots, resulting in noisy conceptual edits. To this end, we verify that *explanations are human-dependable*, i.e. a human is the final evaluator of any explanation, and while a method is able to provide semantically meaningful explanations (in this case relationship edits), it is possible that at the same time the same method provides meaningless explanations (in this case concept edits). Nevertheless, if the derived explanations are not conceptual, a human cannot verify their validity; therefore, we can safely claim that *human interpretability of explanations is highly tied to semantics*.

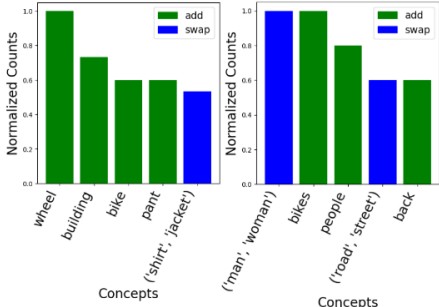

Figure 24: Concepts inserted/ deleted/substituted to implement the 'pedestrian' → 'driver' transition.

**Action Genome**  We test our method in a real-world image dataset extracted from Action Genome (Ji et al., 2020), a video database depicting human-object relations and actions. It is completely unannotated and also like VG has no predetermined classes for its instances. AG results are not presented in the main paper because they offer no new insights compared to other extendability experiments. However, a brief qualitative analysis was deemed interesting enough to present in the appendix. We select a subset of 300 individual frames and generate scene graphs following well-established SGG methods[2]. After applying our CE method using predictions made by Zhou et al. (2017), we obtain results comparable to previous experiments. Specifically, the binary retrieval metrics ranged from 0.17 - 0.41 for P@k and 0.21 - 0.35 for NDCG@k, while overall average edits were 12.86. This experiment validates the relative ease of obtaining graphs from images and demonstrates the applicability of our method to AI-generated graphs of varying quality.

In Fig. 25, we offer some qualitative results on the AG dataset. Instances in this custom AG subset are individual video frames that depict mostly indoor spaces with or without people at a variety of angles and settings. Due to the lack of control in this case, we have identified specific categories that are more meaningful to human perception, such as 'kitchen', 'hall', and 'living room'.

By observing these examples, we can initially note that automatically generated graphs provide a satisfactory representation of the images. However, there are missing details and known biases resulting from imbalanced triple and relation distributions in VG, where the SGG models are trained. We analyze the counterfactuals while acknowledging the potentially lower quality of the input graphs. Since this part of the experiments aims to demonstrate the applicability of our method to unannotated

---

[2]SGG on Action Genome

datasets, in-depth analysis is not performed. Nonetheless, we can observe that the retrieved graphs exhibit structural similarities and share common concepts, which is also visually apparent. For instance, images featuring kitchens often involve the removal of cabinets located above counters, while tables are prevalent in hallway depictions.

## G APPLICABILITY ON OTHER MODALITIES

The process of SMARTY graph generation differs compared to our previous experiments in a few key ways. In this new approach, each user or patient was directly connected to their symptoms and characteristics, which were defined to be audible to a certain extent. Symptom analysis involved treating certain symptoms as sub-symptoms when necessary, based on the hierarchical structure presented in Dervakos et al. (2023)'s SMARTY hierarchy, as opposed to using WordNet for computing node edit costs. Regarding edges in the graph, a simpler strategy was adopted due to the limited number of edge types. Specifically, the approach considered edge swaps between different edge types, as well as the addition and deletion of edges, as costly operations. To initialize the GNN similarity component, custom BioBert (Lee et al., 2020) embeddings were utilized because the language used in the medical field is specific and distinct from general language, unlike previous approaches that relied on simple Glove embeddings. These changes were made to enhance the accuracy and relevance of the SMARTY graph generation.

In Tables 19, 20 comprehensive global edit lists can be found. It is important to note that in Table 19, triple edits refer to edge edits and the concepts adjacent to them. For the sake of readability, we have omitted the head and predicate of the triples, where all heads are the 'User' concept and all predicates represent symptoms or sub-symptoms. Table 20, on the other hand, focuses on node edits, regardless of edges. Evidently, there is agreement with Table 19, but there are also additional noteworthy findings. One of these findings relates to the reported gender bias mentioned in Dervakos et al. (2023), and another suggests a correlation between COVID-19 positivity and younger users.

Table 19: Global triple edits for COVID-19 Negative → Positive.

| Triple Edits | Norm. Counts |
|---|---|
| 'Sneezing' | 1.0 |
| 'RunnyNose' | 0.78 |
| 'DryThroat' | 0.35 |
| 'Fever' | 0.34 |
| 'Dizziness' | 0.31 |
| 'Fatigue' | 0.22 |
| 'Respiratory' | 0.22 |
| 'DryCough' | 0.21 |
| 'TasteLoss' | 0.21 |
| 'Cough' | 0.16 |

Table 20: Global concept edits for COVID-19 Negative → Positive.

| Concept Edits | Norm. Counts |
|---|---|
| 'Sneezing' | 1.0 |
| 'RunnyNose' | 0.73 |
| ('Male', 'Female') | 0.68 |
| 'DryThroat' | 0.36 |
| 'Fever' | 0.35 |
| 'Dizziness' | 0.31 |
| ('Fourties', 'Twenties') | 0.29 |
| 'DryCough' | 0.23 |
| 'Fatigue' | 0.23 |
| 'Respiratory' | 0.23 |

## H LIMITATIONS

Our work is subject to certain limitations. First of all, our experiments involving the CUB and VG datasets are highly dependent on the existing annotations, thus influencing the quality of the derived conceptual explanations. Specifically, the generated semantics through SGG are influenced by the training datasets, namely VG. This limitation was addressed through the comparison of our method's consistency among two vastly different graph generation methods. Despite the positive results validated by the similar produced global edits, there is much room for exploration in this domain. We plan to engage in this venture in our future research. Moreover, pre-trained image classifiers, such as ResNet50 and Places365 may produce imperfect labels for the images under consideration, which may influence the resulting counterfactual explanations. CEs are also characterized by known limitations, such as robustness (Slack et al., 2021). While we have not addressed this particular limitation in our work, we plan to explore it in our future work. Despite these limitations, we have ensured actionability guarantees with the aim of improving the quality of the provided counterfactuals.

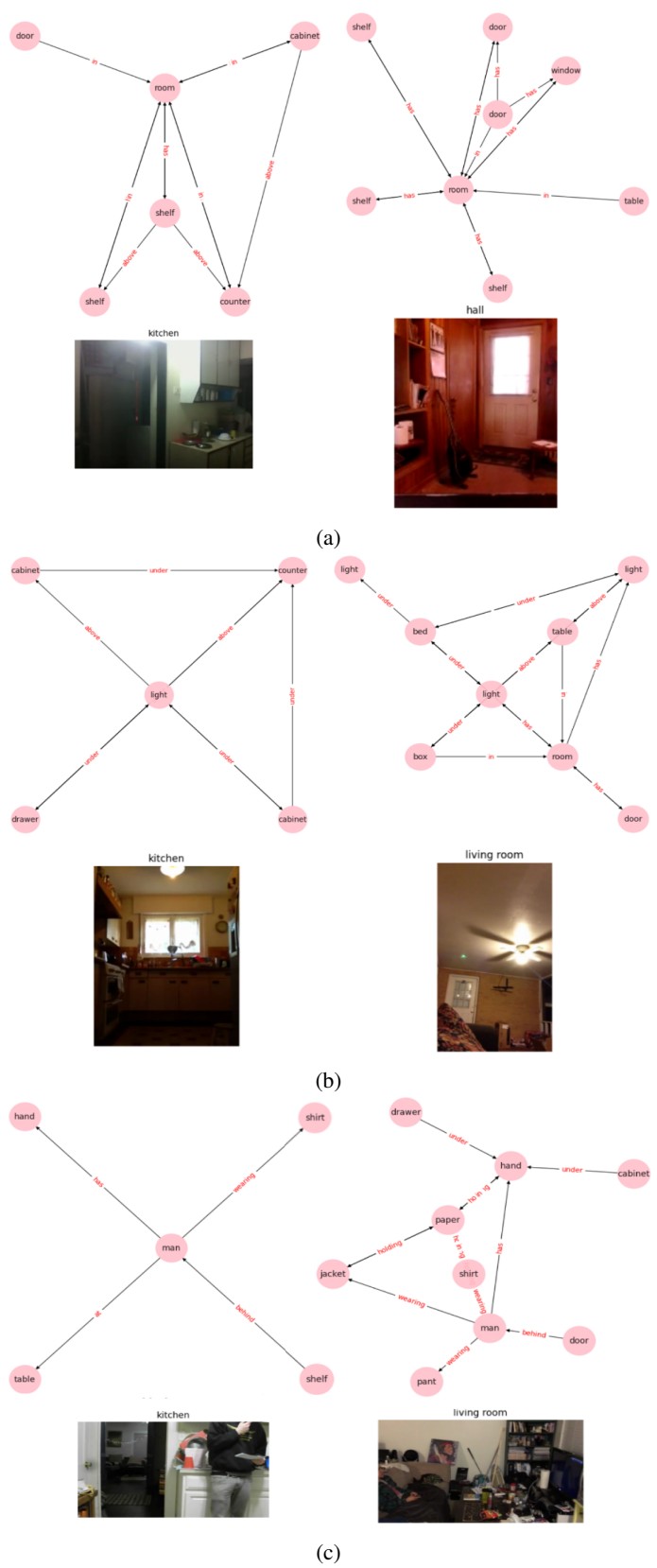

Figure 25: Counterfactual examples from AG dataset for query images belonging to the class "kitchen". Here, CEs are classified as "hall" or "living room".

