# OpenReview forum: "Conceptual Graph Counterfactuals"
_ICLR.cc/2024/Conference — Submitted to ICLR 2024_

### Official Review · Reviewer_wCZ7 · 2023-11-01

**Soundness:** 3 good
**Presentation:** 3 good
**Contribution:** 2 fair
**Rating:** 6
**Confidence:** 3

**Summary:**

The paper proposes to use semantic graphs as a vehicle for Counterfactual Explanations. Since handling distance over graph could be computationally too expensive the paper proposes a novel method based on GNNs to efficiently compute counterfactuals retrieved from a given database. The proposed method is validated both quantitatively and qualitatively on diverse modalities.

**Strengths:**

1. The idea of applying GNNs over scene graphs to compute an approximate graph edit distance for Counterfactual Explanations is novel and interesting.
2. The proposed method is evaluated both against previous SoTA methods and with human annotations.

**Weaknesses:**

1. The main contribution of the paper is rather empirical, to the best of my knowledge none of the proposed techniques is novel, however their joint application to solve the problem statement is interesting. As such, the impact of the work on the community could be increased by providing a more comprehensive evaluation of a broader set of modalities/domains (the current version of the manuscript mostly covers images and very briefly audio signals).
2. The current version of manuscript works under the assumption that the "retrieval database" (of scene graphs) is “dense enough”. Such an assumption is key, especially when using Counterfactual Explanations for explaining specific failure modes of trained models. However, this assumption might not easily hold in practice. Can the authors comment more on this problem and how their new framework is expected to behave in realistic scenarios? It is not difficult to see that, as the scene graphs increase in dimension, all possible graph insertion/deletion/substitution lead to a combinatorial explosion of possible compatible scene graphs that might be hard to cover with a finite amount of data.
3. Results in Table 5 are quite close, however authors claim that their method exhibits superior performance over the baseline, can the authors put the number into perspective? In absolute terms it appears that both models perform quite similarly (as it is highlighted in the qualitative Figure 4 as well).

**Questions:**

Why “computing the GED for only N/2 pairs to contract the training set is adequate for achieving high quality representations? Has it only been validated experimentally or it can be formalized more precisely? Can the authors make this point clearer in the manuscript?


Minor:
- Editorial suggestions:
    - Please define acronyms before use, e.g. in the abstract.
    - The abstract could be summarized without loss of information.
- What is actionability mentioned in the introduction?

---

> ### Author Response · Authors · 2023-11-19
>
> We sincerely appreciate the reviewer's comments and evaluations. We are eager to address any inquiries in the hope of convincingly conveying the contribution of our work, which we believe to be substantial.
>
> Regarding reported weaknesses:
> 1. We wish to draw the reviewer's attention to the novel aspect of our approach, which lies in the representation of the input using semantic graphs. This unique approach enhances the expressivity and flexibility of the proposed counterfactual method, facilitated by the utilization of simple yet effective and efficient GNNs —a combination unprecedented in this field. While it is accurate to acknowledge that each individual technical method employed in our paper is not introduced here, we maintain that the collective innovation stemming from their integration significantly contributes to the overall novelty of our approach. To this end, as described at the end of the Introduction and presented in detail later in the paper, we have undertaken a comprehensive series of experiments encompassing multiple datasets and classifiers. These experiments employ a diverse range of evaluation methods, incorporating both quantitative metrics and human evaluations. Considering the extensive volume of experiments conducted, we assert that showcasing the method's efficacy on two modalities is presently sufficient. Expanding this work further poses challenges in terms of space and readability, given the substantial amount of information already presented in the paper.
>
> 2. Assuming we have correctly interpreted the reviewer's comment, we would like to clarify that the density of the database is not a specific aspect addressed in this work. While we acknowledge the importance of this consideration, we believe that this assumption is not directly tied to our proposed approach but rather represents a general challenge for any counterfactual method in terms of utility and interpretability. The density of our graph database aligns with the data distribution of the input, making it a characteristic inherent to the dataset itself.
> In terms of graph size, it is essential to note that employing a GNN - which does not engage in a matching process in terms of edit operations - significantly mitigates computational demands. The potential "combinatorial explosion" is primarily associated with the computation of supervision GED labels. In our practical experience, even when dealing with larger CUB graphs (Table 9), computation times remained within a reasonable range (Table 8). It is crucial to emphasize that, given our focus on visual classifier explanations, and in alignment with the methods compared, scene graphs predominantly fall within the category of medium-sized graphs.
> 3. Evaluation of explanation methods is a rather challenging task; thus, we have employed several techniques to prove the superiority of our method. Regarding Table 5, we acknowledge the proximity of the reported results for the average number of edits metric, a point that we ourselves have highlighted. However, according to our other comparative measures, we can deduce the superiority in terms of semantic content. Firstly, qualitative results paired with provided analysis prove the importance of structure and relations (‘on’ predicate to separate the surfing activity from simply walking on the beach, correlated toppings etc.). This observation is also reinforced by Section 4.3’s driver vs pedestrian experiment. Secondly, ranking results in terms of the GED gold-standard are consistently higher than SC. To further validate these points, we have added the actual GED between source and counterfactuals in Figure 4 and overall results in Section D.2 of the Appendix. We kindly urge the reviewer to take these into consideration as well.

---

> ### Author Response · Authors · 2023-11-19
>
> Questions:
> 1. The choice of utilizing N/2 pairs for the successful training of the GNN is indeed an experimental observation. This is clearly stated in the manuscript during the description of the proposed method (in Section 3 “Ground Truth Construction” it says “computing GED for only N/2 pairs to construct the training set is adequate for achieving high quality representations, as validated experimentally”). We believe that this empirical result is sound. The conducted study was consistent among several different datasets as well as node embedding components (Tables 6, 15). On top of this ablative result, it is worth noting that employing this specific number of pairs for all reported best scores resulted in superior performance compared to our competitors. This outcome further supports the efficacy of our chosen approach in achieving competitive results while optimizing computational efficiency.
> 2. Actionability, being a concept well-known within the XAI community, is deliberately introduced as a smaller component of our analysis. In the Introduction section, our focus is primarily on establishing motivation, and we intentionally refrain from providing a definition at that stage. Instead, a dedicated and detailed explanation is deferred to the concluding paragraph of Section 4.1.
>
> We trust that our responses have addressed the reviewer's inquiries and concerns. We encourage a reassessment of their evaluations, expressing our gratitude once again for their time and critiques.

---

### Official Review · Reviewer_iSZ3 · 2023-11-01

**Soundness:** 2 fair
**Presentation:** 3 good
**Contribution:** 2 fair
**Rating:** 5
**Confidence:** 3

**Summary:**

The article presents an approach for providing more interpretable Counterfactual Explanations using concept graphs. Assuming instances to be represented as scene graphs and given a target "counterfactual" class, finding a counterfactual explanation is formulated as retrieving the instance of that class with minimum Graph Edit Distance (GED) from the current input. To avoid the costly process of explicitly computing GED for each pair of instances in the dataset, the method proposes to learn to embed scene graphs in such a way that their distance in the embedding space is similar to the GED of their graphs. This objective is supervised by computing GED on a subset of the dataset and experiments show that it generalizes even when fractions of the original dataset are provided. Comparisons with other semantically interpretable counterfactual XAI methods show that the approach retrieves counterfactual images with lower/more coherent GED distance w.r.t. to the ground-truth one.

**Strengths:**

1. Using distances among scene graphs to define counterfactuals is sound and of practical interest: as scene graphs ground the input to semantically interpretable concepts, the distance, and edits are inherently interpretable by humans, as also confirmed via the user studies Table 3 and Table 4.
2. The introduction clearly motivates the proposed approach and provides an extensive overview of Counterfactual Explanation methods and their interpretability issues, motivating the targeted scenario and the employed solution.
3. The approach is flexible as it can be applied to any input representable via graphs, as shown in the audio experiments of Table 7.
4. As graphs are grounded to input instances, the edits are all actionable, a fundamental property for counterfactual explanations.

**Weaknesses:**

1. Most of the quantitative metrics are based on graph distances. For instance, Table 1, and Table 5 report the performance as the average number of graph edits between the retrieved instance by the proposed framework and the competitors. Similarly, Table 2 compares the ranking of the retrieved instance vs the "gold-standard" ranking provided by the ground-truth minimum GED. Given that the main contribution of the work is using GED over scene graphs as a way to provide counterfactuals and that the model is supervised with ground-truth GEDs over scene graphs (while the competitors are not), these results are not completely fair as the metric is biased toward the proposed model. Finding more method-agnostic metrics (as done in Tables 3 and 4) would make the claims stronger.
2. Despite being biased on the metrics, it seems that the competitor SC can still retrieve samples with low edits (Table 5, Section 4.2) achieving either superior or comparable results with the proposed method. Section 4.2 claims the contrary (i.e. superiority of the proposed approach) but the quantitative evidence is not clear. This statement should be refined and the reason for these comparable results expanded.
3. The SC approach (Dervakos et al., 2023) also uses GED but over knowledge graphs. From the text (and Section 2), it is unclear whether the contribution w.r.t. SC is mostly on the type of input graphs rather than on the selection criterion per se. Note that this would also impact the contribution in Section 1, as the article would not be the first to employ graphs for visual counterfactuals (but GNN for fast computation of GED).
4. Tables present inconsistent sets of baselines as Tab. 3 does not report the CVE baseline, and Table 4 misses the SC one. This makes it hard to assess whether the proposed approach is superior to all competitors in all user studies.
5. While using GED over scene graphs it is an interesting direction, it has two potential drawbacks. The first is that concepts of interest should be included in the pretrained scene-graph extractor. The article does not describe at the moment how the performance of the approach varies w.r.t. the scene graph extractor and Sections 4.1 and 4.2 assume graphs to be available (either via automatic extraction from labels or the dataset itself).
6. The second point is that GED may not correlate with distance in the classifier space but rather in the graph-based input representation. This is taken into account methodologically when the retrieved sample is conditioned on the classifier scores ranking (last part of Section 3). Nevertheless, It would be helpful to add a discussion on how the two relate to each other, and even quantitative analyses  (e.g. on L2 distances in the feature space) to check whether distances in the feature/classifier space correlate with GED.
7. For the analyses on Visual Genome, the approach is using a model pretrained on Places365 as a classifier. However, as the examples in Fig. 4 show, the images may depict a single foreground object rather than a scene. The choice of Places365 as a pretraining classifier rather than other choices (e.g. ImageNet) is not motivated in the text.



-------------------------------------------------------------

**Update post-response phase:**

I thank the authors for the detailed answers, clarifying most of the concerns raised by all reviewers. I increased my score accordingly.  However, I still have some concerns, which I still deem as major:

1. The claim that: "GED is the most method-agnostic way to establish semantic distance because it is directly based on all possible semantic changes both in terms of objects as well as relations" is semantic/dataset dependent and it assumes that the graph describes all relevant semantic relations needed for the classification task. This is questionable, as in some tasks changes in semantics are not easily captured via a semantic graph: a naive example are fine-grained manufacturing dataset (e.g. cars, aircrafts) where recognition of a model exploits other cues (e.g. textual for the brand). Note that I am not questioning that GED is a good way of representing semantic changes, especially when ground-truth graphs are available, but just the assumption that GED is the best way to capture them in general.

2. While SC uses a proxy for GED, the article explicitly focuses on finding the shortest GED, facilitated by the specific scene graph used as input. While these choices are indeed a positive aspect and a contribution of the method, the 2/3 comparison tables with SC in the main paper focus on GED (Table 1, Table 5). SC is not included in Table 2 (where GED is nevertheless used as a criterion)  and Table 4 (with human tests). While I understand that Table 4 is a direct replication of the study in CVE, including SC would make the analysis more comprehensive and strengthen the claim that the method outperforms SC even in other metrics (i.e. not related to GED).

**Questions:**

I believe semantic graphs are a useful tool for providing interpretable and semantically grounded Counterfactual Explanations. At the same time, I have concerns regarding the experimental evaluation and the methodology that I hope the rebuttal could address. Specifically:
1. Is considering GED as ground truth for the rankings fair?
2. What are the main differences (in terms of contribution) w.r.t. SC? And why the performance differences with SC is limited for the Visual Genome experiments?
3. Why the user studies focused on different type of baselines?
4. How do performance/explanations vary w.r.t.the underlying scene graph extractor? E.g. applying SGG to extract graphs Visual Genome and CUB would impact the performance of the model?
5. Do distances on scene graphs correlate with distances on the classifier space?
6. What is the reason behind the choice of Places365 as the pretraining dataset for the results on Visual Genome?

**Details Of Ethics Concerns:**

None.

---

> ### Author Response · Authors · 2023-11-19
>
> We sincerely appreciate the reviewer's thorough analysis and attention to detail regarding our work. We are eager to address any inquiries they may have and are committed to providing comprehensive responses in the hope of convincingly conveying the soundness and contribution of our work, which we hold with confidence.
>
> 1. In our experiments we have made a deliberate effort to maintain a consistent evaluation process to ensure fairness. Given the challenging nature of the evaluation of XAI methods, especially when the methods under comparison define distance on a completely different level (pixels vs semantics), we believe that using GED is a good way to establish a ground-truth (refer to paragraph “Evaluation” Section 4). In fact, having motivated the importance of basing explanations on semantics rather than mere pixels [1, 2], we believe GED is in fact the most method-agnostic way to establish semantic distance because it is directly based on all possible semantic changes both in terms of objects as well as relations. Practically optimizing for GED does not introduce bias into our approach; rather, it is a logical consequence of our emphasis on semantic relevance. To be even more precise, there is no doubt about the fairness of GED when comparing our method with SC. SC is also a conceptual method which models CEs by measuring distance. More precisely the authors are actually using a proxy for GED, which focuses on rolling up the edges into concepts. In terms of comparison with CVE, which is also semantically-driven, we believe that semantic distance between the depicted concepts and the way they interact should be quantifiable. Nonetheless, we have also reported further metrics as the reviewer points out. Human evaluation in this case did not just establish that humans prefer our proposed counterfactuals, they can understand the decision process of the classifier so well through conceptual graph counterfactual edits, that during training they do not even need to look at the images (pixel information).
>
> 2. The main contribution in comparison to SC lies on the input representation. We place an emphasis on semantic graphs because they are able to model the relations between objects combined with the underlying structure. As explained, SC do in fact propose the use of GED as a means of counterfactual computation. However, due to their choice of representation through sets of concepts, they are not able to actually compute it. To this end they roll up edges into concepts and perform an algorithm resembling GED. However, as pointed out in our paper the set representation leads to loss of information (Section 2). In the majority of their experiments they end up not leveraging edge information at all (this includes VG, CUB experiments) and when they do, they are still unable to detect “the multiplicity of objects and their relations” (as stated in our paper and explained with examples in the qualitative results of sections 4.1 and F). The novelty in terms of the computation method (i.e. the use of GNNs for GED approximation to facilitate counterfactuals) is a direct consequence which only adds to our contribution.
> Regarding the closeness of results in Table 5, we would like to point out that minimum number of edits is only one of our proposed metrics and as stated in the paper, one should view the results as a whole and with close attention to qualitative insights which can help us deduce the superiority of our method in terms of semantic content. To this end we have added the actual GED between source and counterfactuals in Figure 4 and further overall results in Section D.2 of the Appendix. Firstly, qualitative results paired with provided analysis prove the importance of structure and relations (‘on’ predicate to separate the surfing activity from simply walking on the beach, correlated toppings etc.). This observation is also reinforced by Section 4.3’s driver vs pedestrian experiment. Secondly, ranking results in terms of the GED gold-standard are consistently higher than SC.
>
> 3. The choice of baselines for the user studies is directly correlated with the purpose of each study. We performed two separate human evaluations. The first with the purpose to establish user preference among explanations and the second to test the understability of our semantic–driven graph-based method in comparison to a method purely based on pixels (detailed in section 4 “Evaluation” and further in Appendix A). To this end, in the first one we compared with both baselines, while in the second, a direct replication of the survey in CVE, we compare only with them. After all, we continue our analysis and dive deeper into the comparison with the method most similar to ours in the section directly following. Also the reviewer incorrectly states that CVE comparison is missing from Table 3.

---

> ### Author Response · Authors · 2023-11-19
>
> 4. In Sections 4.1 and 4.2 we utilized the available graphs because they already existed. To address the potential dependence of explanations on the scene graph generator, we conducted the experiments on Section 4.3. The “Unannotated Datasets” section does not only test the framework’s applicability on datasets were graphs are not originally available, but also tests two independent SGG settings - applying a SOTA SGG and creating graphs from image captions to avoid known SGG biases. The similarity in essence in the average counterfactual triple edits they produce, establishes that even with varying annotations (refer to Table 19 for graph statistics) the important semantic context to transition from one class to the other stays the same. This experiment is very close in nature to the approach proposed by the reviewer (using an SGG on VG, CUB)
> 5. Distances are not correlated to the distances in the classifier space. Although an interesting suggestion, we believe it is not in our best interest to adopt such an approach. The information needed to enforce this constraint would require us to peek inside the classifier and therefore automatically render our method white-box, stripping it from its flexibility and potentially its applicability to different modalities.
> 6. The choice of the Places365 classifier was a conscious decision. While ImageNet classifiers are more widely recognized and researched, they are trained on the ImageNet dataset, which primarily consists of foreground objects. In Visual Genome, the majority of instances depict scenes, providing substantial background. Although some instances focus more on specific objects, they are still situated within a particular environment. In contrast, ImageNet classifiers face challenges with such inputs, as only about 3% of the target classes in the corresponding dataset pertain to broader scenes. The reviewer rightfully noted that this analysis was missing from the manuscript, prompting us to address it by adding a paragraph in Appendix C.
>
> We trust that our responses have addressed the reviewer's inquiries and concerns and strongly believe that these clarifications mitigate the grounds for the paper's outright rejection. We encourage a reassessment of their evaluations, expressing our gratitude once again for their time and insights.
>
> [1] Dervakos, E., Thomas, K., Filandrianos, G., & Stamou, G. (8 2023). Choose your Data Wisely: A Framework for Semantic Counterfactuals. In E. Elkind (Ed.), Proceedings of the Thirty-Second International Joint Conference on Artificial Intelligence, IJCAI-23 (pp. 382–390). doi:10.24963/ijcai.2023/43
> [2] Browne, K., & Swift, B. (2020). Semantics and explanation: why counterfactual explanations produce adversarial examples in deep neural networks. arXiv preprint arXiv:2012.10076.

---

### Official Review · Reviewer_B8to · 2023-11-01

**Soundness:** 3 good
**Presentation:** 3 good
**Contribution:** 2 fair
**Rating:** 5
**Confidence:** 4

**Summary:**

This paper propose a novel counterfactual approach that focuses on semantics shifts instead of explaining models' decisions via pixel-level changes. The authors represent images as semantic graphs derived from semantically annotated datasets. Subsequently, the counterfactuals for a given query image are defined as the closest graph (measured by graph edit distance) from another class. To enhance computational efficiency, the authors propose training a GNN to approximate the GED computation. Experimental results on widely-used datasets demonstrate superior performance compared to previous methods.

**Strengths:**

1. The authors employ graph similarity between a query and a target image, which is the first attempt in this direction. The authors have evaluate the method's performance across various datasets spanning different modalities, including CUB (an image classification dataset) and COVID-19 (an audio classification dataset), demonstrating the model-agnostic nature.
2. The authors provide code for reproducibility check.
3. The paper is well-written and easy to follow.

**Weaknesses:**

[Major]

1. **Method & motivation:** A primary objective of counterfactual explanation is to discover why and how the deep model (system) decision changes when specific regions within the given query image are modified. However, the semantics constructed in this paper rely on annotations from the dataset rather than capturing from the target model. Consequently, this raises concerns regarding the fidelity of the explanations generated by the proposed conceptual graph counterfactual method to the deep model's decision-making process. For instance, in the context of adversarial attacks [1-2], imperceptible image modifications to humans can significantly impact the model's output, a phenomenon unaddressed by the proposed method.


>[1] DeepFool: a simple and accurate method to fool deep neural networks. (CVPR 2016)
>
>[2] One Pixel Attack for Fooling Deep Neural Networks. (IEEE Transactions on Evolutionary Computation 2019)



[Minor]

1. The font size in the figures is excessively small, making them particularly challenging to decipher when printed. Furthermore, it is advisable for the authors to employ vector graphics to enhance the quality of the illustrations.
2. As the authors pointed out in Appendix H, the method relies on massive annotation. At present, large vision (multi-modal) models [3-5] have the capacity to produce annotations with rich descriptions. It is advisable to evaluate the proposed method in conjunction with these large models.

>[3] A Unified Model for Vision, Language, and Multi-Modal Tasks. (ICLR 2023)
>
>[4] Learning transferable visual models from natural language supervision. (ICML 2021)
>
>[5] Segment anything. (arXiv preprint 2023)

**Questions:**

My questions are listed in "Weaknesses" section.

---

> ### Author Response · Authors · 2023-11-19
>
> We sincerely appreciate the reviewer’s critiques and would like to directly address their concerns, adhering to their format.
>
> [Major]
> 1. **Method & motivation**: The reviewer's apprehension regarding the proposed framework's dependence on annotations rather than directly capturing the target classifier is essentially a critique of black-box models in general. However, this concern is not a specific issue to be addressed in the context of our paper. Previous work [1] has successfully embraced the black-box setting. As demonstrated in Tables 1, 2, 3, and 4, black-box models like ours not only compare favorably to methods with direct access to the model but can actually outperform them. Their superiority is not only quantitatively substantiated but also validated through human perception—an aspect of utmost significance given that explanations, including counterfactuals, ultimately target human understanding.
> It is essential to note that dismissing our approach based on potential adversarial attack risks may be unjust, as none of the other SOTA approaches provide similar guarantees. Even CVE, which is a white-box pixel-level technique and might initially be perceived as more robust against malicious input changes, fails to explore this specific use case and lacks corresponding insights. It is crucial to emphasize that a comprehensive robustness analysis of our method is a planned aspect of our future work, as outlined in Sections 5 and H. We appreciate the reviewer for these valuable suggestions that will contribute to our ongoing efforts in this direction.
>
> [Minor]
>
> 1. **Figures and Diagrams**: Following this reviewer’s comment, we plan to apply this process for a potential camera-ready version.
> Reliance on annotations:
> 2. As highlighted by the reviewer, our method hinges on the availability or creation of annotations. We have conscientiously addressed this aspect not only in Appendix H but also in Section 4.3 (Unannotated Datasets) of the main paper, along with additional clarification provided in Section F of the Appendix. In our experimental setup, we employed a SOTA scene graph generator and BLIP, one of the most widely used captioners, in conjunction with a well-established relation extraction module to extract annotations. It's worth noting that the large multi-modal models proposed by the reviewer are designed to perform tasks that are not directly correlated with the annotations required by our framework (i.e. image segmentation, VQA, object classification etc.). Given the nature of our framework, where this specific aspect serves as a subsection demonstrating its extensibility, we contend that an exhaustive analysis of annotation construction is beyond the scope of this paper. We acknowledge the potential for constructing intricate pipelines to generate optimal annotations using the proposed models, and we recognize this as an intriguing future exploration.
>
> We trust that our responses have addressed the reviewers' inquiries, encouraging a reassessment of their evaluations. We express our gratitude once again for their time and thoughtful suggestions.
>
> [1] Dervakos, E., Thomas, K., Filandrianos, G., & Stamou, G. (8 2023). Choose your Data Wisely: A Framework for Semantic Counterfactuals. In E. Elkind (Ed.), Proceedings of the Thirty-Second International Joint Conference on Artificial Intelligence, IJCAI-23 (pp. 382–390). doi:10.24963/ijcai.2023/43

---

### Official Review · Reviewer_DyGD · 2023-11-01

**Soundness:** 3 good
**Presentation:** 3 good
**Contribution:** 2 fair
**Rating:** 5
**Confidence:** 2

**Summary:**

This work analyzes scene graphs built on semantic attributes of the data. In particular given a scene graph, this work proposes a method to identify the closest scene graph -- in terms of graph edit distance --- with a different label. Solving this problem is NP-hard and hence, this work adopts an approximate solution that uses a GNN. The GNN computes an embedding for each graph such that the distance between the embeddings of two graphs approximates the graph edit distance. The authors evaluate this method on a variety of datasets.

**Strengths:**

This works designs a novel solution to compute the closest scene graph with a different label. The work uses an simple and elegant solution to the problem that is similar to multi-dimensional scaling (MDS).

The experiments highlight that their work empirically outperforms prior works like CVE and SC. Their proposed method finds graphs with smaller edit distance across different benchmarks. The human evaluators also prefer CEs and find them significantly easier to use them distinguish classes. The author also show that their method is significantly more efficient compared to computing the underlying GED.

**Weaknesses:**

**Understanding the significance of this work: aren't counterfactual explanations used to understand model predictions?**
I am not an expert in this area so I am unable to accurately evaluate the significance of this work. I am unable to understand how these counterfactual explanations will be used. What kinds of insights can they provide and in what scenarios can they be used?

The counterfactual explanations in this work are model-agnostic which means that they cannot be used to understand  how models make predictions. However, I am only familiar with counterfactual explanations (see [1]) that help us understand why models have arrived at a particular decision. Can model-agnostic counterfactual explanations be used for other tasks?

**How do you generate the scene graph for each data point if it isn't available?**
Each data points requires a scene graph which may not be readily available. For example, CUB does not have a ground-truth scene graph and the authors are forced to construct one. As a result, the counterfactual explanations will change depending on how the graph is constructed. If this is the case, how should the graph be constructed in order to get the "right" counterfactual explanation? In this case, is modelling the scene graph the right thing to do?

Also, why is the graph edit distance a reliable way to measure the distance? Perhaps, editing should be assigned less weight compared to deletion when computing the distance.

**Quality of the model is limited by the quality of the labelled data.**
The labelled data uses an approximate algorithm to find the edit distance. As a result, the neural network will also approximate the sub-optimal solution and not predict the optimal answer. Are there ways to get around this hurdle?

**Why use cosine distance in Eqn 4, and why not use euclidean distance instead?**
Since the training objective is also based on the euclidean distance, wouldn't be a better distance measure to compute the nearest scene graph?


**References**

[1] https://jolt.law.harvard.edu/assets/articlePDFs/v31/Counterfactual-Explanations-without-Opening-the-Black-Box-Sandra-Wachter-et-al.pd

**Questions:**

1. **Why do we need a neural network? Why can't we just compute the ground-truth?** Isn't it possible to compute the ground-truth graph edit distance between all pairs of graphs instead of training a neural network to make these predictions. Since we need to create a lot of training data, why can't we compute graph edit distances for the entire data by throwing lots of compute at the problem?
2. **What is error from ground-truth?** How different are the predictions of the network when compared to the ground-truth? Do the feature vectors accurately reconstruct the MDS or are they only correlated? While, the proposed method outperforms other methods, how far away is it from ground-truth?
3. **What are the benefit of being model-agnostic?** What kind of problems can these counterfactual explanations be used to tackle?
4. **What should we do if the scene graph is not readily available for a problem?** How should we construct such a scene graph from semantic attributes?
5. **Why is the counterfactual explanation given by the minimum graph edit distance useful?** Should we count the edit of two different attributes to be equivalent (say stripe pattern and color) or does it make more sense to count them differently.

---

> ### Author Response · Authors · 2023-11-19
>
> We sincerely appreciate the reviewer’s comments and would like to directly address their concerns.
>
> Firstly, we believe that this reviewer has missed or misconstrued some parts of the paper’s motivation and methodology. For instance, the focus of the paper does not simply lie on “analyzing scene graphs built on semantic attributes of the data” or finding “the closest scene graph with a different label”. Omitting the explainability aspect from the analysis greatly shifts the reader’s perspective of this work and undermines its contribution to the field of counterfactuals in terms of flexibility, efficiency and expressiveness.
>
> Some of the reviewer’s suggestions, although very insightful, are in fact already implemented in our approach (“editing should be assigned less weight compared to deletion when computing distance”, counting the distance between attributes differently according to their semantics). As explained in Section 3 subsection Ground Truth Construction, we have employed techniques from SC which consider the WordNet hierarchy for operation cost computation and define deletion and insertion as more costly operations. This information is considered implicitly since GED scores are our supervision signals and it would be impossible to directly employ them for GNNs because they do not perform these operations in the first place. Furthermore, regarding the comment questioning the use of cosine similarity, we would like to highlight that this metric is a straightforward choice for embedding comparison (graph in this case) and preferred here to avoid the curse of dimensionality.
>
> As for the reviewer’s questions, we believe that to an extent they have been covered in the paper. Therefore, in the following response we will point the reviewer to the corresponding sections and provide further clarifications that hopefully answer their concerns.
>
> 1. **Why computation of the ground-truth w/o NNs is prohibitive**: As outlined in the paper, “GED does belong in the NP-hard complexity class”, rendering the computation of the ground-truth not only challenging but practically infeasible for larger graphs (see CUB statistics Table 9 in the Appendix). Even by leveraging an approximation algorithm over the GNN, the calculation process requires several hours (see Table 8). The GNN, as demonstrated empirically and documented, effectively cuts the computation time in half within the proposed transductive setting.
> 2. **Approximation error**: The error is quantified by measuring the deviation from the golden-standard GED ranking, a metric explicitly detailed in the paper. Tables 2, 6 of the main paper present various ranking metrics concerning the retrieval of the ground-truth across different datasets and GNN variants. It's crucial to note that, in terms of the approximation of the similarity score itself, this is, in fact, irrelevant to our approach. Naturally, there exists a trade-off between achieving perfection and computational efficiency. Given the highly demanding computational nature of GED, accepting a margin of error becomes a necessity. This acceptance further underscores the justification for involving human evaluators in validating the retrieved results.
> 3. **Model-agnostic approach**: We believe that there has been a misunderstanding of the term model-agnostic often used in conjunction with the term black-box. To elaborate, "model-agnostic" typically refers to XAI techniques that can be applied to different machine learning models, regardless of the specific algorithm or architecture. The statement “they cannot be used to understand how models make predictions” is therefore false, as further reinforced by the cited paper which explains how black-box CEs can be used to explain the decision-making process for GDPR purposes in Section V.A. To this end, there are no restrictive use cases for this type of explanation. In contrast, model-agnostic CEs offer benefits like increased flexibility which in this case is aided by the graph representation allowing the application of this approach for practically any modality (Section 4.3). If the reviewer’s concern lies on whether a model-specific / white-box technique would be a preferable choice, we refer them to Tables 1,2,3,4 proving the superiority of black-box techniques like ours against a SOTA approach with direct access to the classifiers.

---

> ### Author Response · Authors · 2023-11-19
>
> 4. **Scene Graph Extraction**: The paper addresses the event of the unavailability of annotations in Sections 4.3 and even further elaborates in Section F of the Appendix. In summary, graphs can be extracted via a plethora of available techniques such as Scene Graph Generation, Image Captioning plus Relation parsing. As for CUB, it is mentioned that it has “structured annotations” (Section 4.1 and Table 18 for example); thus, the choice of the representation is heavily guided by the available data.
> 5. As established in the beginning of this response, the fifth question has been already covered in the paper and solved by methods proposed in SC and adopted by the authors.
>
> We trust that our responses have clarified potential points of confusion and addressed the reviewers' inquiries, encouraging a reassessment of their evaluations. We express our gratitude once again for their time and insights.

---

### Author Response · Authors · 2023-11-19
**General Comments**

First of all, we would like to sincerely thank all reviewers for their detailed comments and critiques. We intend to incorporate them into our work, irrespective of the current outcome.

Given the recurring emphasis on the method's reliance on annotations, it is crucial to clarify to all reviewers that this study specifically addresses datasets with semantic annotations, whether in the form of scene graphs or other semantic structures. These types of annotated datasets are increasingly prevalent, and our work not only addresses them but also demonstrates its extendability to datasets lacking such annotations. More than that, exploring the granularity and extraction process of annotations remains an important part of conceptual counterfactuals. Directly quoting from Appendix F of our paper: “an intrinsic characteristic of semantic explanations is their dependence on the knowledge of the individual that provides them. This inherent explainability attribute impacts systems in the same way it does humans. The knowledge supplied to an explainer will determine the specificity and scope of the explanations. Selecting the appropriate annotation technique is a critical step in receiving the desired breadth and depth of explanations.” This aspect has also been addressed in previous work, such as SC, which motivates users to “choose their data wisely”. Explanations are meant for humans and should be guided by human choice.

Granting users the option to select the most fitting semantic description is a pivotal aspect of our approach and is considered a significant asset. In essence, our framework empowers users with the freedom to craft more effective explanations while also assigning them the responsibility to choose descriptions that align with their expertise (i.e. a doctor using a medical audio classifier, as illustrated in our example, might opt for audible symptoms vs. a developer might employ audio features represented as a semantic graph). In line with the CVE approach, the user should carefully select an auxiliary model capable of appropriately encoding the semantic information of the image. The quality of the auxiliary model plays a crucial role in CVE, much like the quality of the semantic description in the proposed method.

To this end, we believe that our framework’s flexibility in that aspect paves the way for future endeavors in retrieving optimal annotations. It should be noted that in the case of CUB, the constructed semantic graphs play a vital role in user understability. Our counterfactual graph explanations allow human evaluators to go one step further from understanding the model’s decisions allowing them to replicate the decision-making process even by only looking at the graph edit CEs without any visual guidance (refer to blind experiment Section 4 “Evaluation” and subsection 4.1 “Human Evaluation”).

Furthermore, we would like to let the reviewers know that the following changes have been made to the manuscript. We have added:
- justification for the choice of Places classifier in Appendix C
- GED for qualitative results (Figures 3, 4)
- average GED for top-1 retrieved counterfactuals (Appendix D.2)
- comments on GED vs number of edits in the main paper and mostly Appendix D.2

---

### Meta-Review · Area_Chair_f5Te · 2023-12-06

**Metareview:**

This paper proposes a new approach to provide interpretable counterfactual explanations using concept graphs. The idea is interesting, and the paper is clearly organized. Extensive results are also provided in the paper. Reviewers raised many concerns regarding the motivation, problem settings, evaluations, etc. Although the authors have addressed some of these concerns in their detailed responses, during the discussion stage, reviewers still found that several major issues remain unclear, such as the GED-based evaluation protocols. Therefore, this paper in its current version is not ready for publication at ICLR.

**Justification For Why Not Higher Score:**

The paper presents an interesting idea for post hoc explainability, but the current version still has some limitations. For instance, the GED-based evaluation protocol is not sufficiently justified.

**Justification For Why Not Lower Score:**

N/A

---

### Decision · Program_Chairs · 2024-01-16

Reject